# Hypoxia enhances antibody-dependent dengue virus infection

Esther Shuyi Gan[1], Wei Fun Cheong[2], Kuan Rong Chan[1], Eugenia Ziying Ong[1] (iD), Xiaoran Chai[3], Hwee Cheng Tan[1], Sujoy Ghosh[3], Markus R Wenk[2] & Eng Eong Ooi[1,4,5,6,*] (iD)

## Abstract

Dengue virus (DENV) has been found to replicate in lymphoid organs such as the lymph nodes, spleen, and liver in post-mortem analysis. These organs are known to have low oxygen levels (~0.5–4.5% $O_2$) due to the vascular anatomy. However, how physiologically low levels of oxygen affect DENV infection via hypoxia-induced changes in the immune response remains unknown. Here, we show that monocytes adapted to 3% $O_2$ show greater susceptibility to antibody-dependent enhancement of DENV infection. Low oxygen level induces HIF1α-dependent upregulation of fragment crystallizable gamma receptor IIA (FcγRIIA) as well as HIF1α-independent alterations in membrane ether lipid concentrations. The increased FcγRIIA expression operates synergistically with altered membrane composition, possibly through increase membrane fluidity, to increase uptake of DENV immune complexes for enhanced infection. Our findings thus indicate that the increased viral burden associated with secondary DENV infection is antibody-dependent but hypoxia-induced and suggest a role for targeting hypoxia-induced factors for anti-dengue therapy.

**Keywords** antibody-dependent enhancement; cellular lipids; dengue; Fc receptors; hypoxia
**Subject Categories** Microbiology, Virology & Host Pathogen Interaction
The EMBO Journal (2017) 36: 1348–1363

## Introduction

Dengue is a mosquito-borne viral infection that affects approximately 390 million people worldwide annually (Bhatt *et al*, 2013). There is currently no therapy for dengue virus (DENV) and the recently licensed vaccine provides incomplete protection against the four antigenically distinct DENV serotypes (Sabchareon *et al*, 2012; Capeding *et al*, 2014; Villar *et al*, 2015). Infection by any of these four serotypes (DENV-1–4) leads to a wide spectrum of clinical outcome that includes severe circulatory shock, internal hemorrhage, and organ dysfunction (World Health Organization, 2009). If not properly managed, mortality rates can be as high as 30% in severe dengue patients (Nimmannitya, 1997; Ooi *et al*, 2006).

Severe disease has been associated with secondary infection, where non-neutralizing antibodies from prior infection with any of the four DENV serotypes bind to the remaining three heterologous DENV. Fragment crystallizable gamma receptor (FcγR)-mediated uptake of these non-neutralized viral immune complexes result in viral entry that, along with other host (Yang *et al*, 2001; Chareonsirisuthigul *et al*, 2007; Halstead *et al*, 2010; Ubol *et al*, 2010; Khor *et al*, 2011) and viral factors (Quiner *et al*, 2014; Manokaran *et al*, 2015), contributes to increased risk of severe dengue. This is uniquely demonstrated in infants, born to dengue-immune mothers, who develop severe disease during a primary infection as a result of sub-neutralizing levels of maternal antibodies that enhance infection (Kliks *et al*, 1988; Simmons *et al*, 2007). Antibody-dependent enhancement (ADE) of infection in cells such as monocytes, macrophages, and dendritic cells results in increased pro-inflammatory and possibly toxic effects of viral antigens that exacerbate disease (Beatty *et al*, 2015; Modhiran *et al*, 2015). Indeed, ADE has recently been demonstrated clinically where cross-reactive antibodies were shown to enhance live attenuated yellow fever virus vaccination infection and corresponding pro-inflammatory response for improved immunogenicity (Chan *et al*, 2016).

Post-mortem analyses have found evidence of active DENV replication in myeloid-derived cells in the lymph nodes, spleen, and liver (Nisalak *et al*, 1970; Basílio-de-Oliveira *et al*, 2005; Balsitis *et al*, 2009; Aye *et al*, 2014). In mice, infection of monocytes and macrophages in lymph nodes is an early event as DENV is trafficked from the site of inoculation by dendritic cells where subsequent rounds of infections occur (Prestwood *et al*, 2012). *In vivo*, the oxygen microenvironments of these lymphoid organs (~0.5–4.5% $O_2$) are substantially lower than atmospheric $O_2$ levels

1 Programme in Emerging Infectious Diseases, Duke-NUS Medical School, Singapore, Singapore
2 Department of Biochemistry, National University of Singapore, Singapore, Singapore
3 Program in Cardiovascular & Metabolic Disorders and Centre for Computational Biology, Duke-NUS Medical School, Singapore, Singapore
4 Department of Microbiology and Immunology, National University of Singapore, Singapore, Singapore
5 Saw Swee Hock School of Public Health, National University of Singapore, Singapore, Singapore
6 Infectious Diseases Interdisciplinary Research Group, Singapore MIT Alliance Research and Technology, CREATE Campus, Singapore, Singapore
*Corresponding author. Tel: +65 65167410; Fax: +65 62212529; E-mail: engeong.ooi@duke-nus.edu.sg

 

(~20% $O_2$) (Caldwell *et al*, 2001; Carreau *et al*, 2011). Low oxygen levels are known to result in differential expression of a number of genes, including those that regulate the inflammatory and immunoregulatory responses in monocytes (Bosco *et al*, 2006, 2011). How such hypoxia-induced modifications in host response affect DENV infection or ADE remain unknown, as mechanistic studies to date have been exclusively performed at atmospheric $O_2$ levels.

In this study, we demonstrate that low oxygen levels induce FcγRIIA protein abundance via hypoxia-inducible factor 1 alpha (HIF1α), which possibly acts as a transcriptional enhancer. With increased FcγRIIA expression, internalization of DENV immune complexes is increased and higher concentration of antibodies are required for complete DENV neutralization. Furthermore, under hypoxic conditions, infection with DENV opsonized with enhancing levels of antibodies resulted in increase in production of infectious DENV progenies. While HIF1α stabilization by hypoxia can explain increased FcγRIIA expression, it alone is insufficient for enhanced infection as chemical stabilization of HIF1α in cells cultured at atmospheric $O_2$ levels showed increased FcγRIIA expression but not viral entry. Instead, hypoxia-induced but HIF1α-independent change in membrane lipid compositions is also necessary for increased FcγRIIA-mediated uptake of DENV immune complexes, which is a key first step for antibody-enhanced DENV infection.

# Results

### Hypoxia upregulates FcγRIIA expression

Low oxygen tensions have been shown, in human primary monocytes, to upregulate the transcription of FcγR, which play a vital role in phagocytosis of antibody-opsonized pathogens (Bosco *et al*, 2006). However, with general inhibition and selective translation under hypoxic conditions, not all upregulated mRNAs necessarily results in changes in protein abundance in the cell (Pettersen *et al*, 1986; Koritzinsky *et al*, 2006; Young *et al*, 2008). To investigate how hypoxia affects the expression of FcγRs on monocytes, we first determined whether primary monocytes cultured at 20 and 3% $O_2$ (herein referred to as normoxia and hypoxia, respectively) for 24 h exhibited changes in FcγR expression. To ensure that data obtained from cell lines can be representative of an *ex vivo* system, response to hypoxia was tested in an acute monocytic leukemia cell line (THP-1) and primary monocytes. Both approaches showed the expected increase in hypoxia-induced genes such as adrenomedullin (ADM) and vascular endothelial growth factor (VEGF) after 24 h of adaptation to hypoxia (Fig EV1 and Appendix Table S1). Consistent with previously reported observations (Bosco *et al*, 2006), exposure to hypoxia increased FcγRI, FcγRIIA, and FcγRIIB mRNA levels as early as 6 h post-oxygen exposure and persisted at least up to 24 h post-oxygen exposure in primary monocytes (Fig 1A–C). However, only FcγRIIA but not FcγRI or FcγRIIB showed increased amounts of protein expression 24 h post-hypoxia exposure as assessed by both Western blot (Fig 1D and E) and flow cytometry (Fig 1F and G, and Appendix Fig S1). Similarly, both the increase in FcγRIIA mRNA and protein expression 24 h post-hypoxia adaptation was observed in THP-1 cells (Fig 1H–J).

### Hypoxia enhances antibody-dependent DENV infections

To determine the effect of hypoxia on the susceptibility of monocytic cells to DENV infections, primary monocytes were infected with DENV only or DENV opsonized with varying concentrations of DENV-2-specific human–mouse chimeric 3H5 (Hanson *et al*, 2006) (h3H5) or pan-serotype human–mouse chimeric 4G2 (h4G2) monoclonal antibody (mAb) (Hanson *et al*, 2006; Chan *et al*, 2011, 2014; Robinson *et al*, 2015). Hypoxic cells required more antibodies for neutralization as compared to cells cultured under normoxic conditions (Fig 2A and B). In addition, plaque titers following DENV-only infection or infection with enhancing concentrations of h3H5 or h4G2 mAbs under hypoxic conditions were higher relative to infection under normoxia (Fig 2A–F). The increase in enhancement was observed with primary monocytes from three other donors (Fig 2D–F) using DENV opsonized with enhancing concentrations of h3H5. However, the increase in plaque titers following DENV-only infection was not universal across all four primary monocyte donors (Fig 2D).

Similar enhancement phenotypes were observed when THP-1 cells were infected with DENV-2 opsonized with h3H5 (Fig 2G), h4G2 (Fig 2H), convalescent sera obtained from individuals previously infected with either DENV-2 (Fig 2I) or DENV-1 (Fig 2J) (Low *et al*, 2006) as well as DENV only (Fig 2K). Increased DENV replication was also observed at lower oxygen concentrations (0.5% $O_2$), suggesting that this phenotype is not constrained to 3% $O_2$ environments (Appendix Fig S2).

### Hypoxia results in increased internalization of antibody-opsonized DENV but not DENV only

The difference in infection rate can be seen as early as 2 hours post-inoculation (hpi) of DENV onto THP-1 cells cultured under hypoxia compared to normoxia (Fig 3A and B). However, treatment with pronase to remove plasma membrane-bound but uninternalized DENV abrogated any differences in RNA levels at 2 hpi for virus only but not antibody-dependent infection (Fig 3C). This suggests that an increase in the internalization of antibody-opsonized DENV under hypoxic condition may be due to the observed upregulation of FcγRIIA in monocytic cells cultured under hypoxic conditions. In contrast, the effect of hypoxia on DENV-only infection is probably mediated through post-viral entry mechanisms.

Next, we determined whether FcγRIIA functionally mediated the observed increase in entry of antibody-opsonized DENV or serves only to aid viral attachment to the plasma membrane (Mady *et al*, 1991; Chotiwan *et al*, 2014; Ayala-Nunez *et al*, 2016). We used high-resolution confocal microscopy and subsequent deconvolution analysis to visualize antibody-opsonized, AF488-labeled DENV-2 with FcγRIIA at resolutions of approximately 200 and 50 nm, respectively (Fig 3D). Using Imaris spot detection and co-localization analysis, spots were placed at center points of DENV-2 and FcγRIIA. This enabled the identification of FcγRIIA interacting with individual antibody-opsonized DENV to within a distance of 100 nm of each other and at a distance of more than 500 nm inside the plasma membrane (Fig 3E). Similar results were obtained when we used ImarisCell to reconstruct the plasma membrane with a highly conservative thickness and identified FcγRIIA-DENV-2 interacting spots within the cytoplasm (Movie EV1). These findings

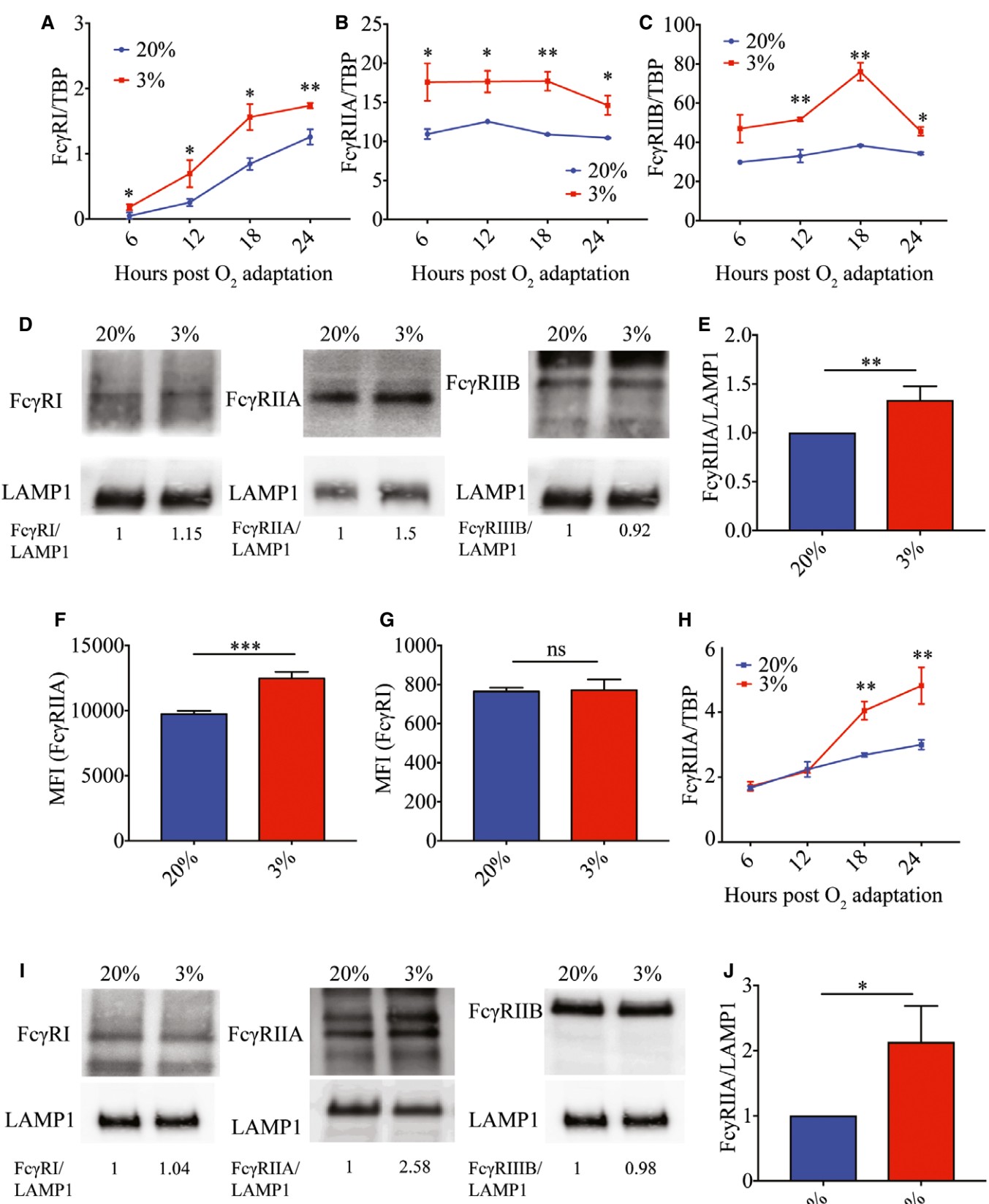

Figure 1.

**Figure 1.  Monocytes express higher levels of FcγRIIA under hypoxic conditions.**

A–C   FcγRI (A), FcγRIIA (B), and FcγRIIB (C) RNA copy number in normoxic (blue) or hypoxic (red) primary monocytes at various time points (hours) post-oxygen adaptation (*n* = 4 biological replicates for each condition).

D   Protein expression of FcγRI, FcγRIIA, and FcγRIIB in normoxic and hypoxic primary monocytes at 24 h post-oxygen adaption. FcγRI and FcγRIIB blots were obtained from the same gel and lanes. The membrane was cut and probed with respective antibodies for FcγRI and FcγRIIB and share the same LAMP1 loading control. Numbers shown below the blots indicate densitometry analysis of protein levels of FcγRs normalized to LAMP1 (data shown is representative of *n* = 4 biological replicates for each condition).

E   Protein abundance of FcγRIIA from four donors in primary monocytes relative to LAMP1 (*n* = 4 biological replicates for each condition).

F, G   Mean fluorescence intensity (MFI) of FcγRIIA (E) and FcγRI (F) in normoxic (blue) and hypoxic (red) primary monocytes at 24 h post-oxygen adaptation as assessed using flow cytometry (*n* = 4 biological replicates for each condition).

H   THP-1 FcγRIIA RNA copy number under normoxic (blue) and hypoxic (red) in THP-1 cells conditions at various time points post-oxygen adaptation (*n* = 4 biological replicates for each condition).

I   Protein expression of FcγRI, FcγRIIA, and FcγRIIB in THP-1 cells at 24 h post-oxygen adaptation. FcγRI and FcγRIIB blots were obtained from the same gel and lanes. The membrane was cut and probed with respective antibodies for FcγRI and FcγRIIB and share the same LAMP1 loading control. Numbers shown below the blots indicate densitometry analysis of protein levels of FcγRs normalized to LAMP1 (data shown is representative of *n* = 4 biological replicates for each condition).

J   Protein abundance of FcγRIIA from four independent experiments in THP-1 cells relative to LAMP1 (*n* = 4 biological replicates for each condition).

Data information: In (A–C, F–H) data are presented as mean ± SD. *P < 0.05, **P < 0.01, ***P < 0.001 (unpaired *t*-test). In (E, J) data are presented as mean ± SEM. *P < 0.05, **P < 0.01 (paired *t*-test).
Source data are available online for this figure.

indicate that FcγRIIA directly mediated uptake of DENV immune complex during antibody-dependent infection under hypoxic conditions.

Concordant to the microscopy data, stable knockdown of FcγRIIA expression using shRNA resulted in significantly decreased DENV plaque titers under hypoxic conditions (Fig 3F).

### HIF1α enhances FcγRIIA expression

The upregulation of FcγRIIA under hypoxic conditions suggest that the transcription factor hypoxia-inducible factor (HIF1α or HIF2α) may play a role in regulating its expression. Under oxygen-rich environments, HIFs are degraded through $O_2$- and iron-dependent reactions (Semenza, 2004). At low $O_2$ or iron levels, HIF binds the core penta-nucleotide sequence, or hypoxia-response element (HRE), to initiate transcription. To evaluate whether the increase in FcγRIIA transcription is mediated by either HIFs, we cultured HIF1α- or HIF2α-silenced THP-1 cells under hypoxic conditions. As a control, we measured for the expected dose-dependent decrease in ADM mRNA expression, which has been previously shown to be regulated by HIFs (Keleg *et al*, 2007). HIF2α-silenced cells (Fig EV2A) showed no changes in FcγRIIA protein expression (Fig EV2B), DENV internalization (Fig EV2C), or DENV replication under ADE conditions (Fig EV2D). However, silencing HIF1α (Fig 4A) resulted in reduced expression of both ADM (Fig 4B) and FcγRIIA (Fig 4C and D).

Likewise, chemically stabilizing HIF1α by an iron chelator desferrioxamine (DFX) under normoxic conditions (Fig 4E) (An *et al*, 1998) resulted in dose-dependent increase in ADM (Fig 4F) as well as both FcγRIIA mRNA (Fig 4G) and protein levels (Fig 4H). This stabilization of HIF1α and subsequent increase in FcγRIIA expression resulted in increased immunofluorescence signal of DENV immune complexes, which was abrogated upon knockdown of FcγRIIA (Fig 4I). These results suggest that FcγRIIA transcription is regulated by HIF1α.

Chromatin immunoprecipitation sequencing (ChIP-seq) analysis, however, did not reveal a HIF1α binding site in the promoter region of FcγRIIA. ChIP-seq could not be carried out at 20% $O_2$ tension due to the rapid degradation of HIF1α (Appendix Table S2). However, under hypoxic conditions, a binding site for HIF1α was identified approximately 5,000 bases upstream of FcγRIIA (Fig 5A). A similar binding site for HIF1α was also observed upon HIF1α stabilization by DFX under normoxic conditions (Fig 5B). These results suggest that HIF1α is not a promoter but potentially an activator of FcγRIIA expression although further work will be needed to confirm the latter notion.

Curiously, however, knockdown of HIF1α and its resultant decrease in FcγRIIA expression only led to a modest decrease in infection of antibody-opsonized DENV (Fig 5C). No difference was observed in virus-only control infection. These findings suggest that additional hypoxia-induced but HIF1α-independent changes may act

**Figure 2.  Hypoxia modulates DENV infection in monocytes.**

A, B   Plaque titers of normoxic (blue) and hypoxic (red) primary monocytes (Donor 1) following infection with DENV-2 opsonized with varying concentrations of h3H5 (A) and h4G2 (B) at MOI of 10 at 72 hpi. Using PRNT (PRNT$_{50}$ < 1:10), Donor 1 was confirmed to have no pre-existing antibodies (*n* = 4 biological replicates for each condition).

C   Plaque titers of normoxic (blue) and hypoxic (red) primary monocytes from Donor 1 following DENV-only infection (*n* = 4 biological replicates for each condition).

D–F   Plaque titers of normoxic (blue) and hypoxic (red) primary monocytes from Donor 2 (D), Donor 3 (E), and Donor 4 (F) when infected with DENV-2 opsonized with 0.391 μg/ml h3H5 that permits maximal levels of ADE. Dashed lines represent plaque titers of normoxic (blue) and hypoxic (red) cells following DENV-only infection in each individual donor (*n* = 4 biological replicates for each condition).

G, H   Plaque titers of normoxic (blue) and hypoxic (red) THP-1 cells when infected with DENV-2 opsonized with varying concentrations of h3H5 (G) and h4G2 (H) (*n* = 4 biological replicates for each condition).

I, J   Plaque titers of normoxic (blue) and hypoxic (red) THP-1 cells when infected with DENV-2 opsonized with different levels of serotype-specific (I) and cross-reactive (J) convalescent serum (*n* = 4 biological replicates for each condition).

K   Plaque titers of normoxic (blue) and hypoxic (red) THP-1 cells following DENV-only infection (*n* = 4 biological replicates for each condition).

Data information: In (A–K) data are presented as mean ± SD. **P < 0.01, ***P < 0.001, ****P < 0.0001 (unpaired *t*-test).

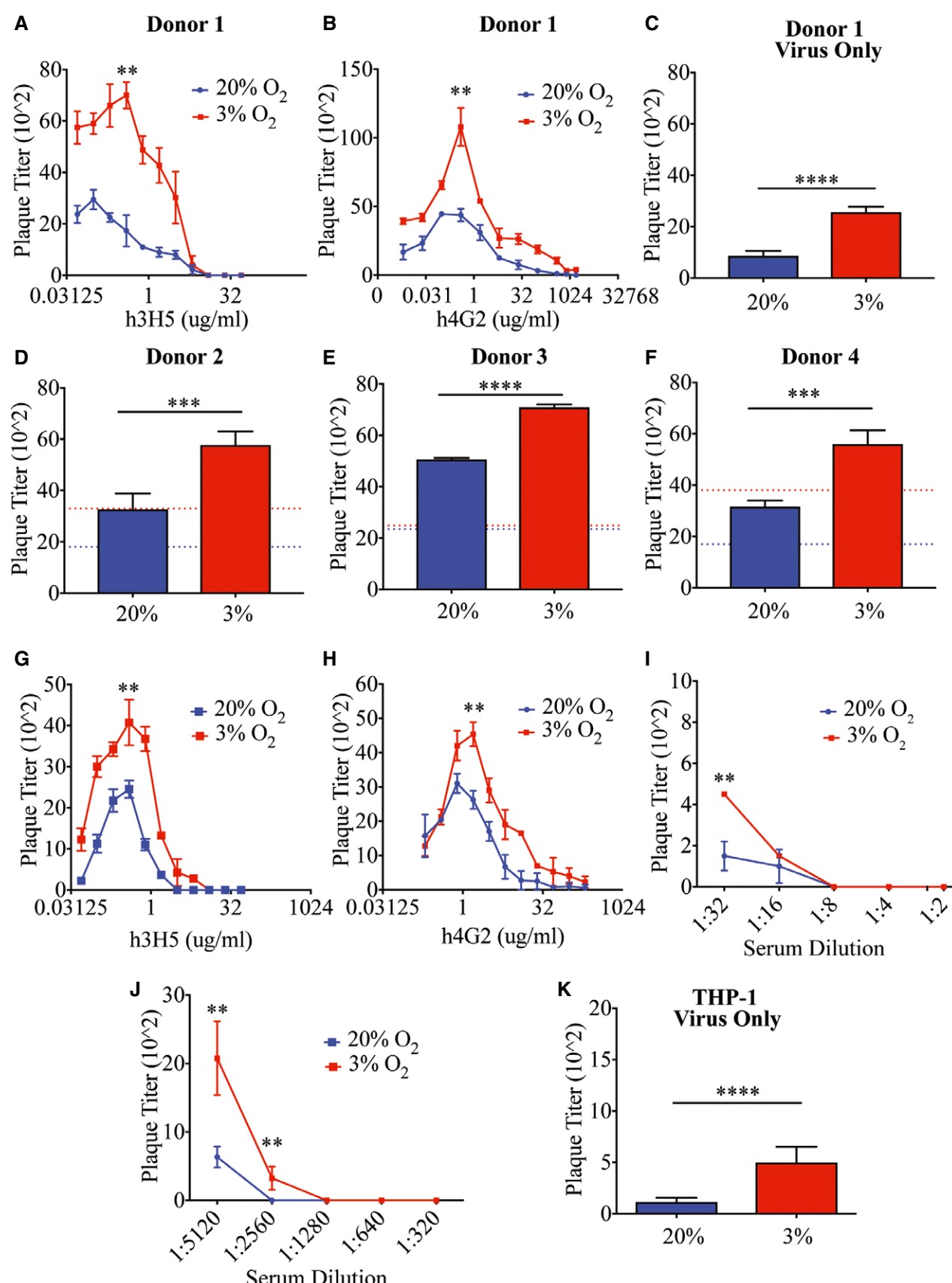

Figure 2.

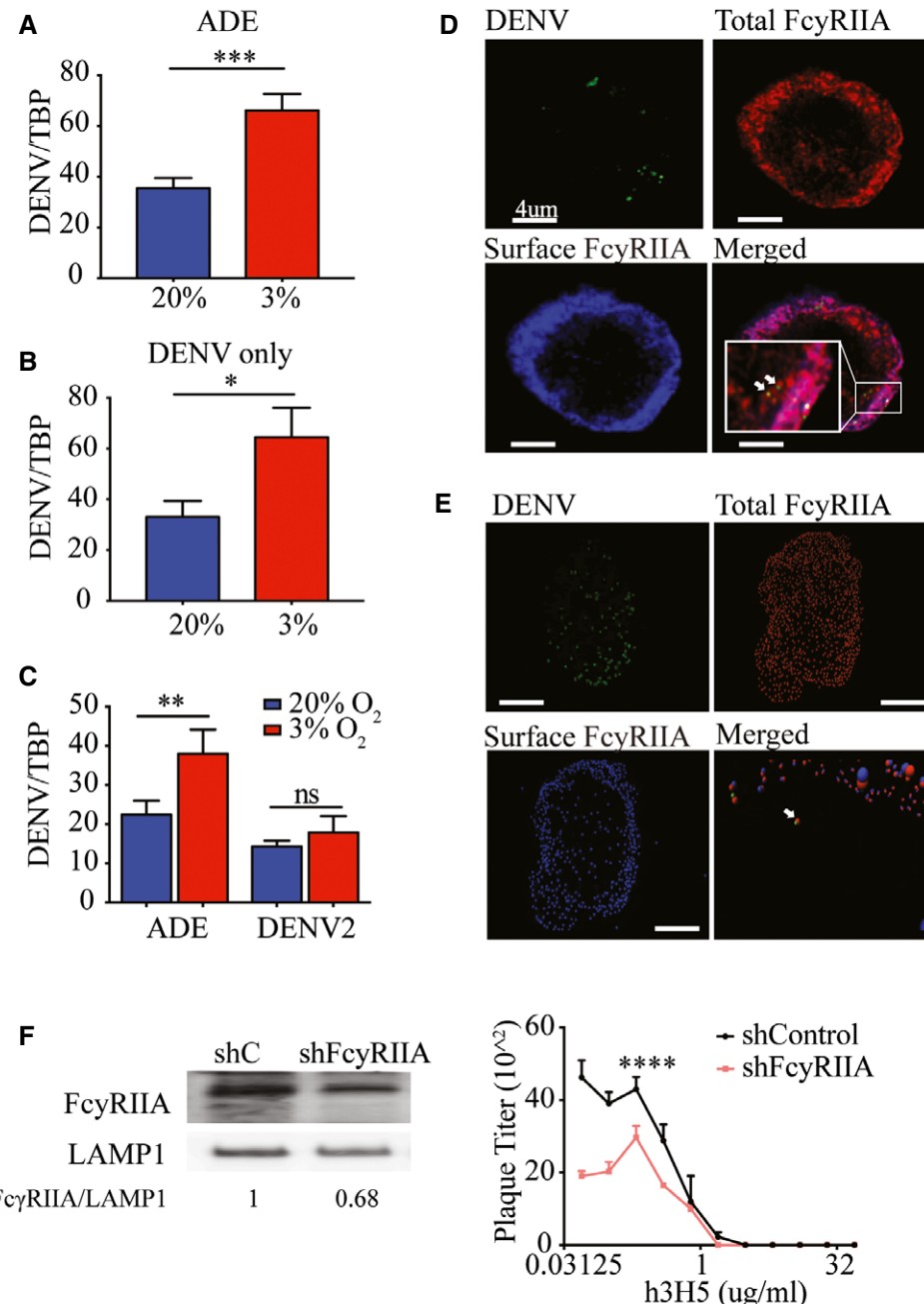

**Figure 3. Hypoxia-induced FcγRIIA expression increases uptake of antibody-opsonized DENV.**

A, B   Viral RNA copy numbers at 2 hpi with DENV-2 opsonized with h3H5 (ADE) (A) or DENV-2 only (B). Blue and red bars represent cells cultured in normoxia and hypoxia, respectively (n = 4 biological replicates for each condition).

C   Viral RNA copy numbers after pronase treatment at 2 hpi under ADE or DENV-2 only conditions (n = 4 biological replicates for each condition).

D   Immunofluorescence microscopy of surface FcγRIIA (blue, AF532), total FcγRIIA (red, AF647) with DENV (green, AF488). Co-localization of internalized FcγRIIA with DENV in the cytoplasm of THP-1 is indicated by the white arrows. Scale bars represent 4 μm.

E   Imaris spot detection of total FcγRIIA (red, AF647), surface FcγRIIA (blue, AF532), and DENV-2 (green, AF488) interaction. Colocalization of red and blue spots represents surface FcγRIIA, whereas red-only spots represent internalized FcγRIIA. Images were acquired at 100× magnification using a Leica TCS SP8 confocal DMI6000 Microscope with hybrid detectors (HyD) and processed using Leica Application Suite X 2.0.1.14392 (LAS X). Deconvolution analysis was performed using Huygens Professional v15.05 and Imaris x64 8.1.2 software. White arrow in the merged image indicates co-localization of DENV (green) with internalized FcγRIIA (red only). This co-localization was identified with an additional 6× magnification and tilting of the image for better visualization, as shown in the accompanying video within the Source Data. Scale bars represent 4 μm.

F   Plaque titers following ADE infection in shControl and shFcγRIIA THP-1 cells at 72 hpi. FcγRIIA silencing efficiency was assessed by Western blot with LAMP1 as a loading control. Numbers under the Western blot indicate levels of FcγRIIA normalized to LAMP1 (n = 4 biological replicates for each condition).

Data information: In (A–C, F), data are presented as mean ± SD. *P < 0.05, **P < 0.01, ***P < 0.001, ****P < 0.0001 (unpaired t-test).
Source data are available online for this figure.

synergistically with FcγRIIA to increase ADE. To further investigate this, we infected hypoxic, normoxic, and normoxic cells treated with 200 μM of DFX simultaneously with antibody-opsonized DENV, with or without pronase treatment. This allows us to ascertain whether other hypoxia-induced but HIF1α-independent factors contribute to the increase in internalization of antibody-opsonized DENV observed in hypoxic monocytes. The results showed increased DENV signal, as measured by flow cytometry, in both hypoxia (51%) and DFX treatment (47%) as compared to normoxic THP-1 cells (25%) (Fig 5D). However, pronase treatment showed that increased uptake of antibody-opsonized DENV remained high only in hypoxic cells (43%) as compared to normoxic cells (20%). In contrast, only 25% of DFX-treated THP-1 cells showed DENV internalization, which is only a marginal increase compared to normoxic, untreated cells. Collectively, these findings indicate that hypoxia-induced internalization of antibody-opsonized DENV requires HIF1α-independent changes in the cell to complement the induced FcγRIIA expression.

### Increased membrane ether-linked PE aids internalization of DENV immune complexes

To identify HIF1α-independent host cell response to hypoxia that could act in concert with FcγRIIA for increased ADE, we profiled mRNA transcripts in THP-1 cells cultured under normoxia and hypoxia for 24 h. Gene set enrichment analyses (GSEA) comparing hypoxic to normoxic cells identified lipid metabolic pathways such as phosphatidylethanolamine (PE), phosphatidylserine (PS), cholesterol, and glycosphingolipid metabolism to be enriched under hypoxic conditions (Fig 6A).

Lipids, especially phospholipids, are important in maintaining cellular membrane integrity and proper function of membrane proteins (Dowhan, 1997). To identify the membrane lipid composition modification that alters the efficiency of FcγR-mediated phagocytosis (Ebbesen et al, 1991), we compared the lipid profile of cells cultured under hypoxia against normoxia. Our results showed that ceramides (Cer), phosphatidylserine (PS), sphingomyelin (SM), phosphatidylinositol (PI) (Fig EV3A–E), phosphatidylcholine (PC), lyso-phosphatidylcholine (LPC), lyso-phosphatidylethanolamine (LPE) (Fig EV4A–D) were not significantly different between hypoxic and normoxic cells. While cholesterol also showed no difference (Fig EV4E), cholesterol esters were marginally upregulated in hypoxic cells (Fig EV4F). Significant differences, however, were observed in a subset of PE lipids. Ether-linked PEs were upregulated up to 4-fold in hypoxic compared to normoxic cells (Fig 6B). In contrast, DFX-treated cells

showed no change in ether PE concentration as compared to normoxic cells (Fig 6C). Furthermore, silencing HIF1α (Fig EV5A) had no significant effect on concentration of PE lipids (Fig EV5B–D), suggesting that upregulation of ether PE is HIF1α independent.

Phosphatidylethanolamine is one of the major structural lipids in eukaryotic membranes and assumes a conical molecular geometry due to the relatively small size of its polar headgroup (Holthuis & Menon, 2014). Any subtle change in PE composition could thus affect cell membrane fluidity (Braverman & Moser, 2012) that alters FcγR-mediated uptake of antibody-opsonized DENV. To test this possibility, we used siRNA to silence alkylglycerone phosphate synthase (AGPS), a requisite enzyme for ether lipid synthesis (Fig 7A) (Benjamin et al, 2013). As expected, reduced AGPS expression significantly reduced uptake of antibody-opsonized DENV (Fig 7B) and the resultant plaque titers produced (Fig 7C) under hypoxic conditions. Furthermore, double knockdown of AGPS and FcγRIIA (Fig 7D) resulted in further decrease in uptake of antibody-opsonized DENV compared to silencing only either AGPS or FcγRIIA (Fig 7E). Collectively, our data indicate that upregulation of ether-linked PE complements the increased FcγRIIA expression to increase susceptibility to antibody-dependent infection under hypoxic conditions.

## Discussion

Monocytes that traffic to lymph nodes and the spleen ultimately function in low $O_2$ environments. While low $O_2$ alters FcγR expression transcriptionally (Bosco et al, 2006), it is unknown how FcγR protein levels and other cellular functions that are affected can impact DENV infection. This study thus provides a first molecular view of the contributory role of low-oxygen environments on cellular functions that impact dengue pathogenesis.

Oxygen level is known to affect viral pathogenesis in different ways. Generally, hypoxia impairs viruses that naturally infect tissues with high atmospheric oxygen concentration such as influenza virus (Magill & Francis, 1936), adenovirus (Pipiya et al, 2005), and simian virus 40 (Riedinger et al, 1999). In contrast, hypoxia has been shown to increase the infection of viruses, such as hepatitis C virus (HCV) (Vassilaki et al, 2013) and sendai virus (Ebbesen et al, 1991), which naturally target organs with low oxygen concentrations such as the liver, spleen, and lymph nodes, as does DENV. We show in this study that hypoxia increases DENV infection through both increased uptake of antibody-opsonized DENV and hitherto undefined post-uptake mechanism for DENV-only inoculation.

**Figure 4.    Increased FcγRIIA expression is modulated by HIF1α.**

A    HIF1α protein levels in control and HIF1α-silenced THP-1 cells. LAMP1 served as a loading control. Numbers indicate levels of HIF1α normalized to LAMP1.

B–D    Expression of ADM (B), FcγRIIA mRNA (C), and FcγRIIA protein (D) 24 h after siRNA control (siC) treatment or knockdown of HIF1α in THP-1 cells under hypoxic conditions. Numbers below the Western blot (D) indicate levels of HIF1α normalized against LAMP1.

E    HIF1α proteins levels in THP-1 cells treated with DFX and cultured at 20 and 3% $O_2$ for 24 h. Numbers indicate levels of HIF1α normalized against LAMP1.

F–H    Levels of ADM (F), FcγRIIA mRNA (G), and protein expression of FcγRIIA (H). All three panels show levels at 24 h post-HIF1α stabilization using media only, 100 or 200 μM DFX in THP-1 cells under normoxic conditions. Numbers below the Western blot (H) indicate levels of HIF1α normalized against LAMP1.

I    Percentage of control or FcγRIIA-silenced THP-1 cells positive for AF488-labeled DENV-2 upon treatment with 100 or 200 μM DFX compared to untreated cells. Proportion of infected cells was measured using flow cytometry at 2 hpi (n = 4 biological replicates for each condition).

Data information: In (B, C, F, G, I) data are presented as mean ± SD. *P < 0.05, **P < 0.01, ***P < 0.001, ****P < 0.0001 (unpaired t-test).
Source data are available online for this figure.

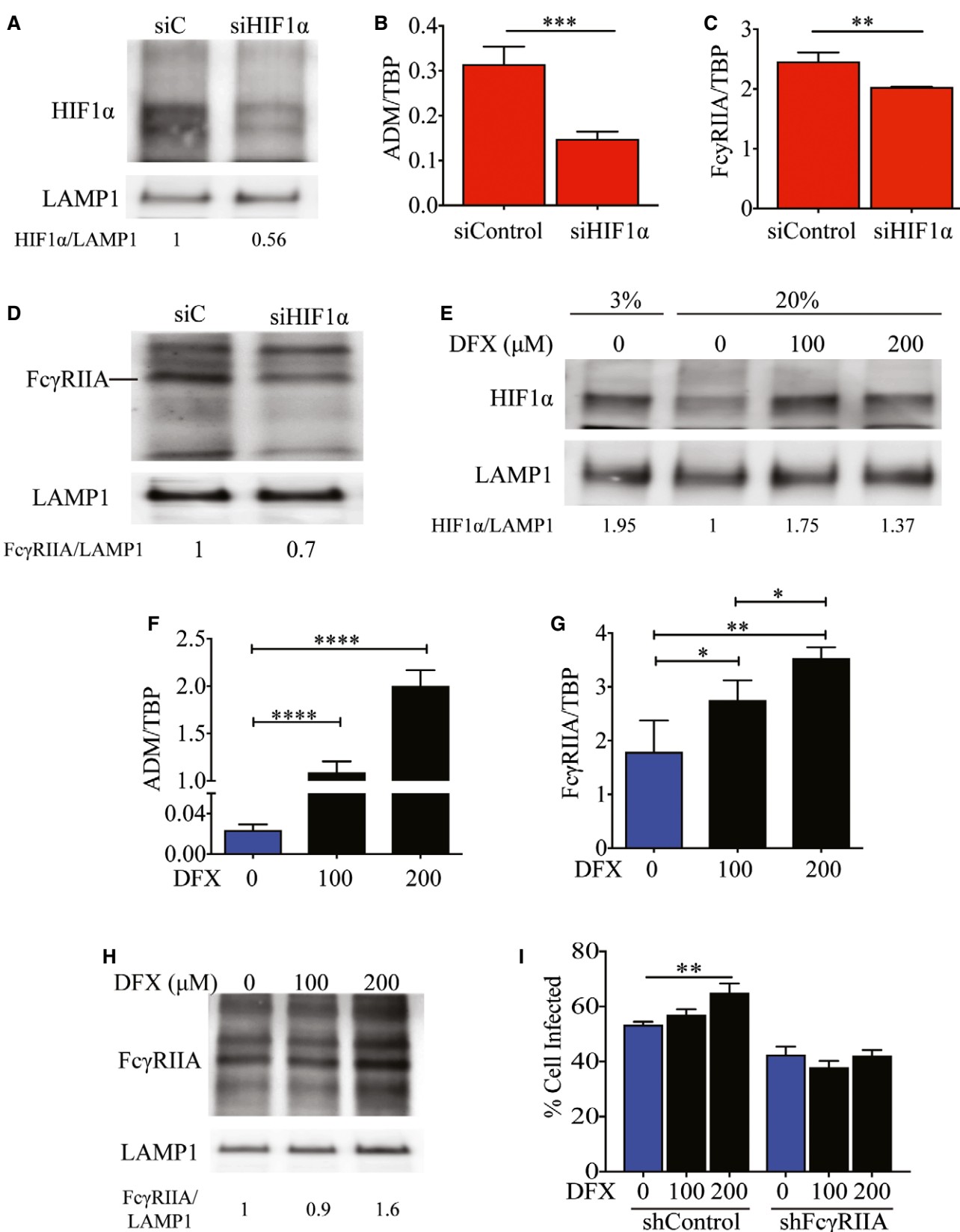

Figure 4.

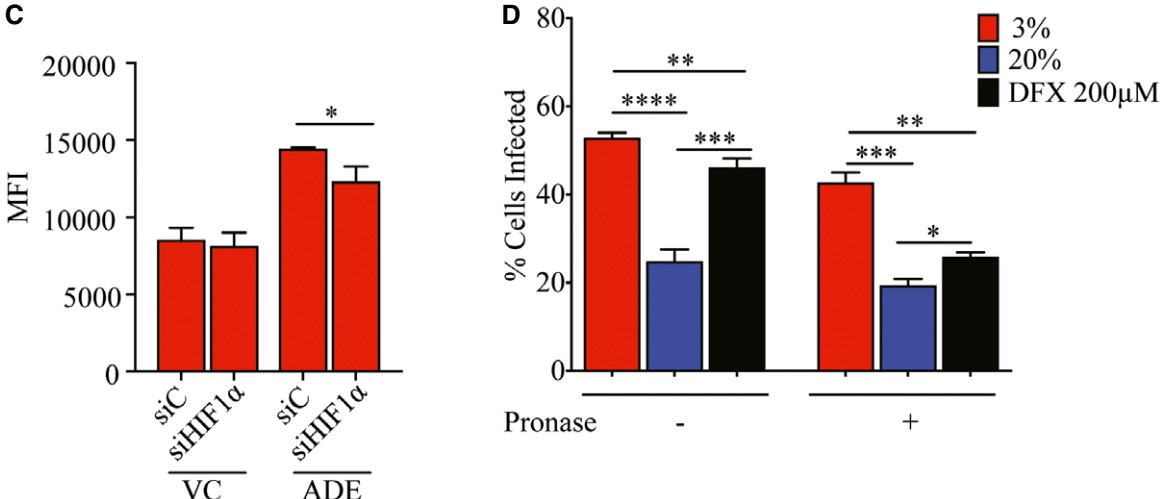

**Figure 5. HIF1α enhances FcγRIIA expression for binding but not internalization of antibody-opsonized DENV.**

A   HIF1α-binding site upstream of the FcγRIIA gene in THP-1 under hypoxic conditions as determined using ChIP-Seq.

B   HIF1α-binding site upstream of the FcγRIIA gene in THP-1 treated with 200 μM DFX under normoxic conditions determined using ChIP-Seq.

C   Control and HIF1α-silenced THP-1 cells cultured under hypoxic conditions and infected with AF488-labeled DENV-2, without or with opsonization with enhancing levels (0.391 μg/ml) of h3H5. Cells were treated with pronase at 2 hpi to assess DENV internalization by MFI (*n* = 4 biological replicates for each condition).

D   Percentage of THP-1 cells infected with AF488-labeled DENV-2 opsonized with enhancing levels of h3H5, without or following pronase treatment. THP-1 cells were cultured under hypoxic conditions or under normoxic conditions without or with 200 μM DFX. Percentage infection was determined by flow cytometry (*n* = 4 biological replicates for each condition).

Data information: In (A, B), HRE sequence in peak region is denoted by #. In (C, D), data are presented as mean ± SD. *P < 0.05, **P < 0.01, ***P < 0.001, ****P < 0.0001 (unpaired *t*-test).

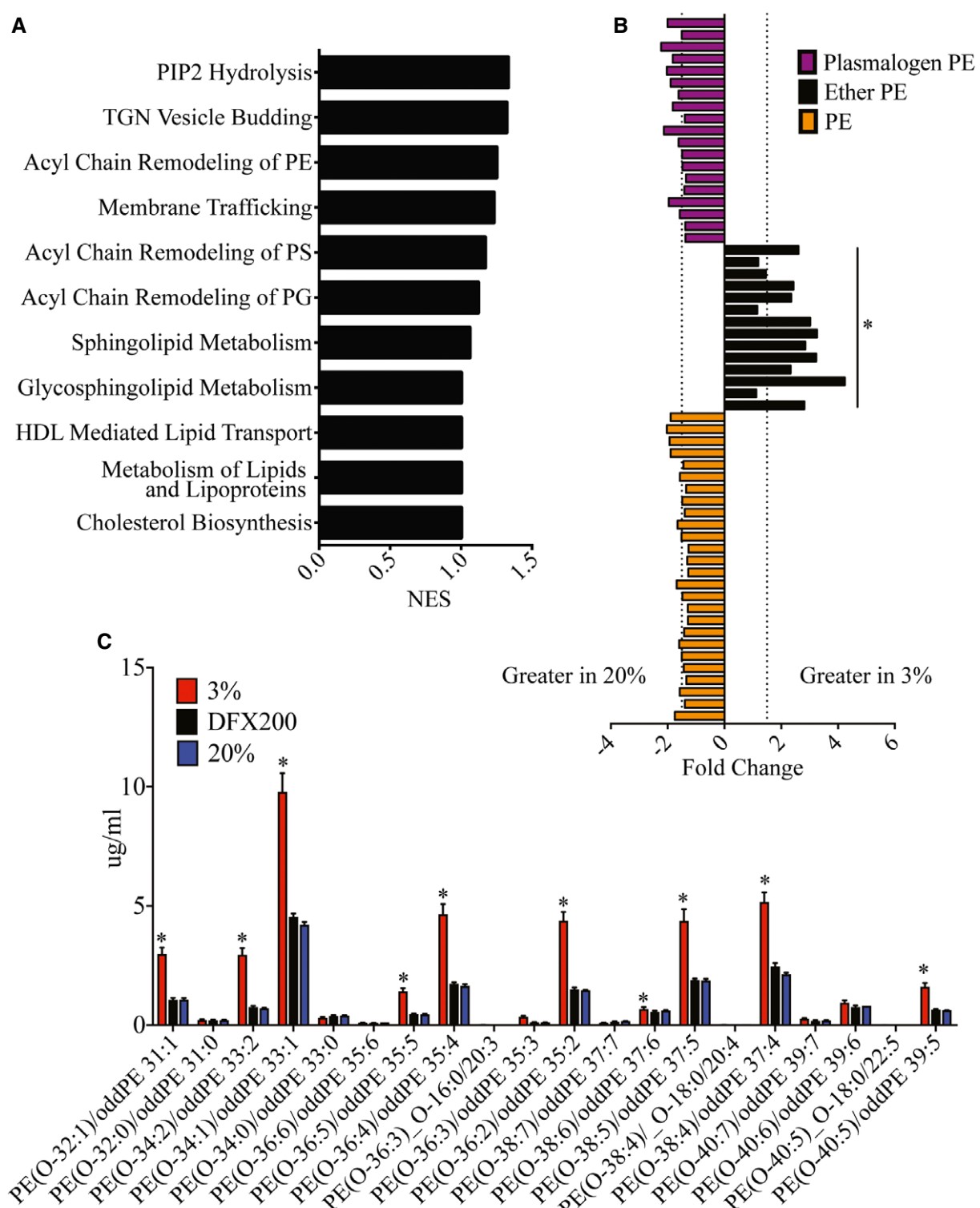

**Figure 6. Hypoxia increases membrane ether PE concentrations independent of HIF1α.**

A   Lipid metabolism pathways enriched in hypoxic THP-1 cells as identified by GSEA analysis.

B   PE concentrations in THP-1 cells following incubation in 3% $O_2$ and 20% $O_2$ for 24 h. Ether PE lipid concentrations increased under hypoxic conditions. No changes were observed in other PE or plasmalogen lipid species. Individual lipid species and fold changes are listed in Appendix Tables S3 and S4. Dotted lines represent a fold change of 1.5 ($n$ = 3 biological replicates for each condition).

C   Ether-linked PE concentrations in hypoxic compared to normoxic DFX-treated (200 μM) THP-1 cells ($n$ = 3 biological replicates for each condition).

Data information: In (B, C) data are presented as mean ± SD. *$P$ < 0.05 (unpaired *t*-test).

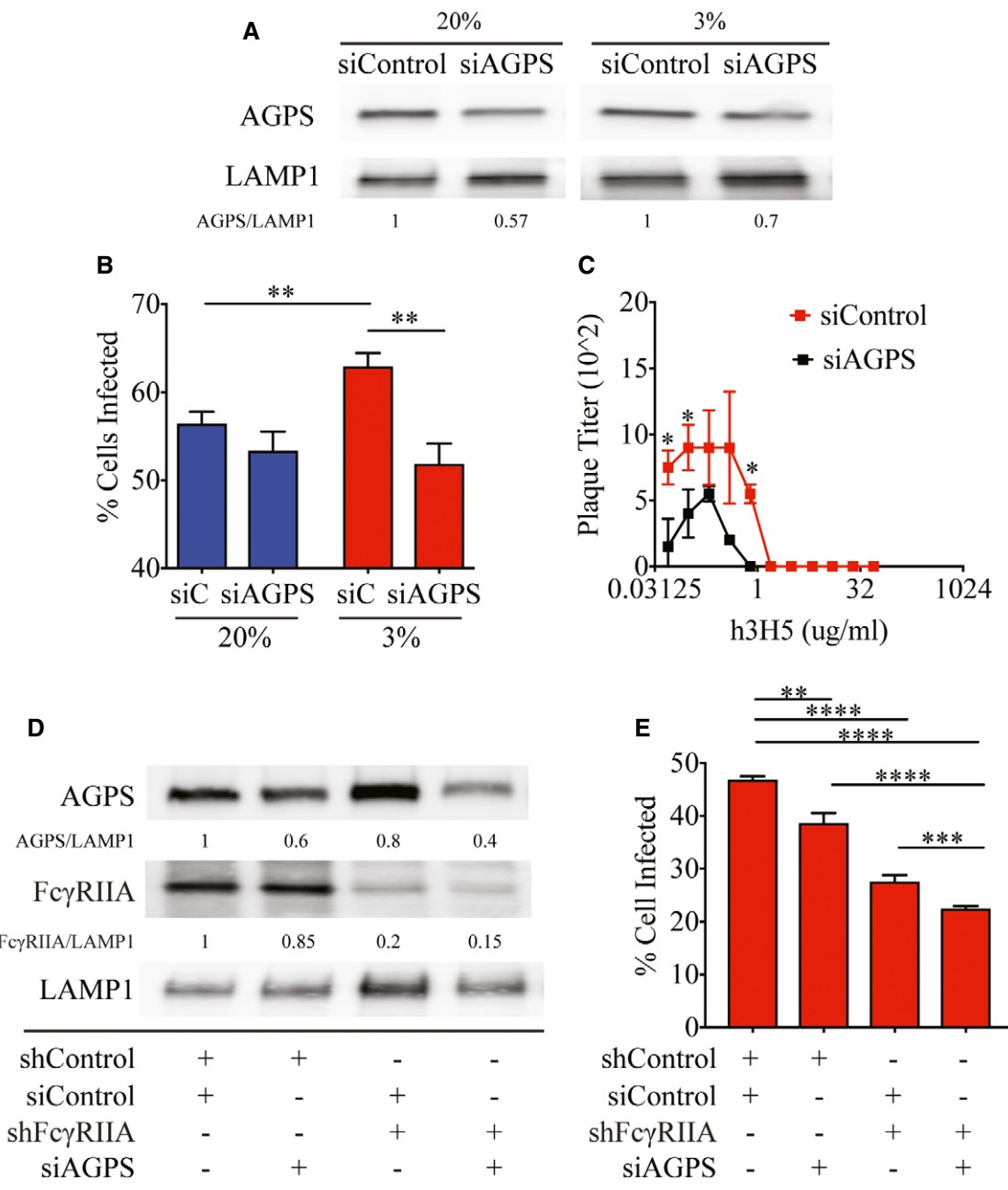

**Figure 7. Upregulation of ether lipids and FcγRIIA synergistically increased uptake of antibody-opsonized DENV.**

A  Knockdown efficiency of AGPS in THP-1 cells cultured at 20 and 3% $O_2$ assessed by Western blot. LAMP1 served as a loading control. Numbers below the Western blot indicate levels of AGPS normalized to LAMP1.

B  Percentage of control and AGPS-silenced THP-1 cells infected under ADE conditions at 2 hpi in 20 and 3% $O_2$ ($n = 4$ biological replicates for each condition).

C  Plaque titers following infection of control (red) and AGPS-silenced (black) THP-1 cells with DENV-2 opsonized with varying concentrations of h3H5 under hypoxic conditions at 72 hpi ($n = 4$ biological replicates for each condition).

D  Knockdown efficiency of AGPS in shControl and shFcγRIIA THP-1 cells cultured at 3% $O_2$ assessed by Western blot. LAMP1 served as a loading control. Numbers under the Western blot indicate levels of AGPS or FcγRIIA normalized to LAMP1.

E  Percentage of AGPS and/or FcγRIIA-silenced THP-1 cells internalized under hypoxic conditions ($n = 4$ biological replicates for each condition).

Data information: In (B, C, E) data are presented as mean ± SD. *$P < 0.05$, **$P < 0.01$, ***$P < 0.001$, ****$P < 0.0001$ (unpaired *t*-test).
Source data are available online for this figure.

Antibody-dependent infection is augmented by hypoxia through FcγRIIA induction, through a possible activator role of HIF1α as well as HIF1α-independent alteration in membrane ether-linked PE

levels. Both changes act synergistically for increased internalization of DENV immune complexes. Indeed, these findings could explain why overexpression of FcγRIIA in cells cultured at atmospheric

oxygen levels had, on some occasions, failed to show increased FcγR-mediated entry (Chotiwan *et al*, 2014). Hypoxia also appears to induce FcγRI and FcγRIIB mRNA levels without detectable differences in the protein levels, suggesting additional regulatory mechanisms on translation by hypoxia that remain to be fully identified. Nonetheless, hypoxia-driven effects on cells targeted by DENV could be an essential factor in dengue pathogenesis, especially in individuals with pre-existing DENV antibodies from a previous infection with a heterologous DENV serotype.

Interestingly, virus internalization during DENV-only infection in the absence of antibodies was not affected by the changes in cellular lipid composition, despite increased replication and production of DENV progeny in both primary monocytes and THP-1 cells. This difference suggests that cellular lipid composition may impact the efficiency of FcγR-mediated phagocytosis differently from other receptor-mediated endocytosis or macropinocytosis, the cellular entry pathway used by DENV-only infection (Ayala-Nunez *et al*, 2016). Additional studies to define how hypoxia affects DENV-only infection will be needed to confirm these possibilities.

Adaptation to hypoxia results in a multitude of changes in the cell, ranging from metabolic pathways (Masson & Ratcliffe, 2014; Eales *et al*, 2016), autophagy (Bellot *et al*, 2009; Heaton & Randall, 2011; Lai *et al*, 2016) to host immune responses (Bosco *et al*, 2006; Nizet & Johnson, 2009). It is clear that hypoxia-induced changes for DENV production remain to be fully elucidated and cellular autophagy, which has previously been shown to enhance DENV replication (Heaton & Randall, 2011; McLean *et al*, 2011), may also be important for antibody-enhanced infection. The observations reported here suggest that these mechanisms could be productive areas of investigation to understand dengue pathogenesis.

Besides pathogenesis, the role of hypoxia in DENV infection also raises other important questions that need to be explored urgently. *In vitro* measurements of neutralizing antibody titers required to confer protection is especially vital to determine vaccine immunogenicity or dose of therapeutic antibodies required against dengue. Our findings suggest that assays conducted at atmospheric oxygen tensions potentially underestimate the amount of antibodies required for complete protection *in vivo* under physiological oxygen tensions due to hypoxia-induced increases in FcγRIIA. Sub-neutralizing levels of antibodies increases the risk of triggering antibody-enhanced infection, which may result in severe disease. Development of assays using cells that express the relevant repertoire of FcγR cultured under hypoxic conditions could provide a fresh layer of information on protective immunity following vaccination.

In conclusion, our findings suggest that the enhanced infection often observed in secondary dengue is antibody dependent and hypoxia induced. Developments in drugs that target hypoxia-induced factors as anti-neoplastic therapy could thus also have antiviral efficacy.

# Materials and Methods

### Primary samples

Primary monocytes were derived from blood obtained from the Singapore Health Sciences Authority Blood Bank, under a protocol approved by the institutional review board (IRB 201406-01). Donor 1 was tested for pre-existing DENV antibodies by PRNT and was found to be negative for antibodies against the four DENV serotypes.

### Cells

THP-1 cells were obtained from ATCC. Primary monocytes were isolated from healthy donors and cultured as described (Chan *et al*, 2011). When required, cells were adapted to 3% $O_2$ in fully humidified incubators supplied with pure nitrogen gas to reduce oxygen levels as well as 5% $CO_2$ for 24 h prior to infection with DENV-2. Cells tested negative for mycoplasma.

### Virus infection

DENV-2 (ST) is a clinical isolate obtained from Singapore General Hospital. Endotoxin-free h3H5 and h4G2 chimeric human/mouse antibodies were constructed as described (Hanson *et al*, 2006). DENV-2 was incubated with media, antibodies, or serum at 37°C for 1 h before inoculation onto THP-1 cells at MOI of 10. Internalization of virus was assessed by Alexa Fluor (AF) 488-labeled DENV 2 hpi as described (Zhang *et al*, 2014) or by qPCR. Viral progenies produced 72 hpi were assessed by plaque assay 72 hpi.

### DFX treatment

Desferrioxamine mesylate salt (DFX) was purchased from Sigma (D9533) and reconstituted in water. DFX (Sigma D9533) was diluted in water to a stock concentration of 500 μM. THP-1 cells were subsequently treated with 100 μM or 200 μM of DFX (diluted in RPMI) for 24 h prior to infection with DENV at MOI of 10.

### Virus assessment with qPCR

RNA from cells was extracted using the RNeasy Kit (Qiagen) followed by cDNA synthesis (Bio-Rad) and real-time qPCR (Roche) according to manufacturer's protocol. Primer sequences were obtained from Origene. DENV primers used were as follows: DENV-F 5′-TTGAGTAAACYRTGCTGCCTGTAGCTC DENV-R 5′-GAGACAG CAGGATCTCTGGTCTYTC. All RNA levels were measured relative to TATA-box binding protein (TBP).

### Plaque assays

Virus plaque titers were quantified as previously described (Chan *et al*, 2011).

### Flow cytometry

After infection, primary monocytes and THP-1 cells were washed with phosphate buffer solution (PBS) and fixed with 3% paraformaldehyde at 4°C. Blocking of FcγRIIA was performed with goat polyclonal CD32A (1:100, R&D Systems AF1875) for 30 min at 4°C. After washing with PBS, mouse anti-CD32 IV.3 (1:300, Stem Cell Technologies 60012) was then added for 30 min at 4°C. After further washing with PBS, anti-mouse AlexaFluor 488 (1:400) was added and incubated at 4°C for 30 min prior to analysis with the BD

LSRFortessa flow cytometer. In a separate experiment, primary monocytes and THP-1 cells cultured at 20% $O_2$ and 3% $O_2$ for 24 h were washed with PBS and fixed with 3% PFA. Mouse anti-FcγRI (1:300, Abcam ab46679) and mouse anti-CD32 IV.3 (1:300, Stem Cell Technologies 60012) antibodies were added for 30 min at 4°C. After washing with PBS, anti-mouse AlexaFluor 488 (1:400) was added and incubated at 4°C for 30 min prior to analysis with the BD LSRFortessa flow cytometer.

## Western blot

Cells were washed once in PBS and resuspended in lysis buffer (1% Nonidet P-40, 150 mM NaCl, 50 mM Tris pH 8.0) in the presence of protease inhibitors (Sigma). Proteins in cell lysates were separated by SDS–PAGE, transferred to PVDF (Millipore) and incubated with primary antibody followed by HRP-conjugated anti-rabbit (1:3,000, Abcam 6721) or anti-mouse (1:1,000, Dako P0447) antiserum. Primary antibodies for LAMP1 (1:1,000, eBioscience 6721), FcγRI (1:300, Abcam ab46679), FcγRIIA (1:300, Stem Cell Technologies 60012), FcγRIIB (1:500, Abcam ab123240), AGPS (1:250, Sigma HPA030209), HIF1α (1:1,000, Abcam 2158), and HIF2α (1:1,000, Abcam 8365) were used. Blots were developed using enhanced chemiluminescence detection reagents (Amersham). Data shown are representative of three independent experiments. Quantification of protein densitometry was performed with ImageJ 1.47v.

## Confocal microscopy

After infection with AF488-labeled DENV and fixation with 3% PFA for 30 min, THP-1 cells were subjected to cytospin at 800 *g* for 3 min and washed with PBS. Mouse anti-CD32A (1:300, Stem Cell technologies 60012) was added and incubated for 1 h at room temperature. After washing with PBS, anti-mouse AF568 (1:200) was added and incubated at room temperature for 45 min. Cells were washed with PBS and permeabilized with 0.1% saponin in 5% BSA for 30 min at room temperature. Anti-CD32A was added for 1 h at room temperature. After washing, anti-mouse AF647 was added and incubated at room temperature for 45 min. Thereafter, cells were fixed with 10% glycerol and 90% PBS before viewing under a Leica confocal microscope. Deconvolution (HuygensEssential) and Imaris analysis were performed at the SingHealth Advanced Bio Imaging Core.

## shRNA and siRNA transfection

siRNA transfection was performed as previously described (Chan *et al*, 2014). siAGPS (s16248), siHIF1α (s6539), and siHIF2α (s4698) from Life Technologies were used. CD32A shRNA (sc-29992-v) was purchased from Santa Cruz Biotechnologies. Qiagen AllStars negative control siRNA (SI03650318) served as control siRNA.

## Microarray

Following RNA extraction, microarray was performed at Duke-NUS Genome Biology Core Facility using Illumina human HT-12 bead-chips. Data normalization was performed using Partek software and quantile normalized prior to analysis with Gene Set Enrichment

Analysis. The microarray data were deposited in the Array Express database (www.ebi.ac.uk/arrayexpress) under accession number E-MTAB-4692.

## ChIP-Seq

Chromatin immunoprecipitation was performed using SimpleChIP Enzymatic Chromatin IP Kit (CST #9003) according to the manufacturer's protocol. After recovery of ChIP and input DNA, 30 ng of DNA was used for each sequencing library preparation (New England Biolabs, E6240); four libraries were multiplexed (New England Biolabs, E7335) and sequenced on one lane of HiSeq 2500 (Illumina) to an average depth of 20–30 million reads per library. The quality of the sequencing reads was ascertained via the FASTQC tool (Li & Dewey, 2011). The average sequencing depth was 54.4 million reads per sample, and the median per-base quality was > 30 for all the samples. No further trimming of the bases was performed. Sequencing reads were then mapped to the human reference genome (hg19) using BWA-0.5.9 alignment tool (Li & Durbin, 2009). The average mapping rate was 99.5%. Peak calling was performed using MACS-1.4.2 (Zhang *et al*, 2008), resulting in the generation of potential HIF1α-binding sites.

## Liquid–liquid extraction

THP-1 membranes were prepared by incubation in 20 mM Tris buffer pH 7.4. Chloroform:methanol (1:2) was added to cell suspension (after washing the cells with 1× PBS, reconstitute cells with 50 μl of MQ water). Known amount of internal standards was spiked into the cell–solvent mixture prior to incubation. The mixture was incubated in a 4°C room for 2 h with agitation at 600 rpm on a Thermomixer (Eppendorf). After incubation, 300 μl of chloroform and 250 μl of MQ water were added into the mixture and vortexed to mix. The mixture was centrifuged at 15,000 *g* for 5 min to break phase and the bottom organic phase which contain lipids was transferred into a clean tube. Re-extraction was performed by using another 500 μl of chloroform, and the organic phase obtained was pooled. The lipid extracts were then dried under nitrogen stream and kept at −80°C until used.

## Lipids analysis using high-performance liquid chromatography/mass spectrometry

Lipids were analyzed on an Agilent 1290 HPLC system coupled with an Agilent 6460 Triple Quadrupole mass spectrometer. Liquid chromatography was performed on a Zorbax Eclipse Plus, Rapid Resolution High Definition, 1.8 μm reversed-phase C18 100 Å, 50 × 2.1 mm column (Agilent Technologies Corp, Santa Clara, CA, USA). HPLC conditions: injection volume 2 μl; mobile phases A and B consisted of isopropanol:acetonitrile in the ratio of (60:40) and (90:10) (optima grade), respectively, both containing 10 mM ammonium formate; flow rate 0.5 ml/min, 60% B for 2 min, then linearly changed to 0.4 ml/min in 2 min; solvent composition was then linearly switched to 100% B in 7 min and maintained for 2 min, and then linearly changed to 20% B in 9 min and maintained for 1.5 min. Then, the flow rate and the mobile phase were returned to the original ratio. Mass spectrometry was recorded under the positive ESI mode. Individual lipid species were quantified using spiked

internal standards obtained from Avanti Polar Lipids (Alabaster, AL, USA).

### Statistical analysis

Experiments were replicated three times (unless stated otherwise), each with four biological replicates. Mean $\pm$ SD of quadruplicate determinations from one representative experiment is shown. Primary monocytes obtained from four different donors were used, with each dataset representing four replicates from a single donor obtained from a single experiment; two-tailed unpaired *t*-test was performed to compare between the means of two conditions of the same donor using GraphPad prism v7.0. A two-tailed paired *t*-test was performed using GraphPad prism v7.0 when multiple donors were included in a single analysis.

**Expanded View** for this article is available online.

### Acknowledgements

We thank Xing Man Jie, Xu Chang, Tanu Chawla, Jolander Si-Hui Lim, and Summer Li-Xin Zhang for their technical assistance, Professors David Silver and Yin Bun Cheung for their guidance on membrane lipids and statistical analyses, respectively. We also thank the anonymous reviewers for their constructive criticisms and very useful suggestions. Confocal imaging and Imaris analysis were performed at the SingHealth Advanced Bioimaging Core. Sequencing was performed at the Duke-NUS Genome Biology Core Facility. This work was supported by the Singapore National Research Foundation under its Clinician-Scientist Award administered by the National Medical Research Council.

### Author contributions

ESG and EEO designed the study. ESG, EZO, HCT performed the *in vitro* experiments. ESG and KRC performed the transcriptomic analysis. WFC and MRW performed the lipid profiling. XC and SG performed the ChiP-Seq analysis. ESG and EEO analyzed the data and wrote the first version of the manuscript. ESG, WFC, MRW and EEO reviewed and revised the manuscript.

### Conflict of interest

The authors declare that they have no conflict of interest.

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
