## [Review Process File · The EMBO Journal]

Manuscript EMBO-2016-95642

Hypoxia enhances antibody-dependent dengue virus infection

Esther Shuyi Gan, Wei Fun Cheong, Kuan Rong Chan, Eugenia Ziying Ong, Xiao Ran Chai, Hwee Cheng Tan, Sujoy Ghosh, Markus R Wenk, Eng Eong Ooi

Corresponding author: Eng Eong Ooi, Duke-NUS Medical School

Review timeline:	Submission date:	01 September 2016
	Editorial Decision:	02 November 2016
	Revision received:	10 January 2017
	Editorial Decision:	06 February 2017
	Revision received:	09 February 2017
	Accepted:	17 February 2017

Editor: Karin Dumstrei

Transaction Report:

1st Editorial Decision

02 November 2016

Thank you for submitting your manuscript to The EMBO Journal. I am sorry for the delay in getting back to you, but I have now received the comments from the three referees.

As you can see below, the referees appreciate the interest of your study. However revisions are also needed in order to consider publication here. The referees raise a number of constructive comments that I would like to ask you to address in a revised version. The comments are clearly outlined below and I will not repeat them all here. In addition it will also be important to address the concern raised by referee #1 in the general comments namely that "that the effect of anoxia on the efficiency of completion of the replication and assembly of dengue viruses was not studied in the presence and absence of enhancing antibodies." This is an important point and we would need such additional analysis for consideration here.

I should add that it is EMBO Journal policy to allow only a single round of major revision and that it is therefore important to address the raised concerns at this stage.

When preparing your letter of response to the referees' comments, please bear in mind that this will form part of the Review Process File, and will therefore be available online to the community. For more details on our Transparent Editorial Process, please visit our website: http://emboj.embopress.org/about#Transparent_Process

Thank you for the opportunity to consider your work for publication. I look forward to your revision.

REFEREE REPORTS

Referee #1:

The authors studied various cell-based phenomena that occur when primary human monocytes or THP-1 cells infected by dengue viruses under enhancing conditions are maintained at low compared with ambient (atmospheric) oxygen concentrations. They observed increased expression of Fc gamma receptors and an increased dengue 2 infection of these cells in the presence of enhancing concentrations of two monoclonal antibodies. The observed increase in growth of dengue 2 virus was attributed to an increase in uptake of infectious immune complexes. Experimental design was restricted, perhaps because of disagreements with results of others, with the result that the effect of anoxia on the efficiency of completion of the replication and assembly of dengue viruses was not studied in the presence and absence of enhancing antibodies. Anoxia may differentially effect the contribution of innate immunity to ADE.

Specific comments:

P 3, line 10. Please do not forget infants who acquire the dengue vascular permeability syndrome as the result of infection enhancement from mothers who have experience two or more dengue infections. ADE does not occur simply as the result of one prior dengue infection.

P 3, line 14. Does the absence of references to other described mechanisms of intrinsic ADE mean the authors reject or simply choose to ignore them? 1,2

P3, line 17. "pro-inflammatory response?" Presumably, the authors do not accept data that supports a direct role of dengue NS1 as a component of tissue damage in acute dengue infections?^{3,4}

P 12, line 3. Who is it that estimates vaccine efficacy as being due to the circulation of dengue antibodies at levels that prevent dengue enhanced infections, in vitro? Please find a sensible rationale for this research.

P 12, Primary samples. In order to attribute infection to enhancement, it is critically important to test all PBMC donors for dengue antibodies and exclude all those who are positive. PBMC from dengue immune donors will support dengue infection in the absence of added antibodies. Donors from antibody positive donors can be used for studies such as those described here, but dengue infection enhancement cannot be attributed to antibodies. Please see 5

P 12, Virus infection. What MOI was used to infect cells? No description is provided of methods for study of enhanced dengue infections of primary human monocytes.

Figure 3. What time after infection was dengue virus growth performance measured? These studies are performed under enhancing conditions but they do not measure impact on infection enhancement per se which requires a non-antibody control.

1. Ubol S, Halstead SB. How Innate Immune Mechanisms Contribute to Antibody-Enhanced Viral Infections. *Clin Vaccine Immunol* 2010; 17(12): 1829-35.
2. Boonnak K, Dambach KM, Donofrio GC, Marovich MA. Cell Type Specificity and Host Genetic Polymorphisms Influence Antibody Dependent Enhancement of Dengue Virus Infection. *J Virol* 2011; 85(4): 1671-83.
3. Beatty RP, Puerta-Guardo H, Killingbeck S, Glasner D, Harris E. Dengue virus non-structural protein 1 triggers endothelial permeability and vascular leak that can be inhibited by anti-NS1 antibodies. *Science Translational Medicine* 2015; 7(304): 304ra141.
4. Modhiran N, Watterson D, Panetta AK, et al. Dengue virus NS1 is a viral toxin that activates cells via TLR4 and disrupts endothelial cell monolayer integrity. *Science Translational Medicine* 2015; 7(304): 304ra142.
5. Halstead SB. Neutralization and antibody dependent enhancement of dengue viruses *Adv Virus Research* 2003; 60: 421-67.

Referee #2:

This study investigates the effects of hypoxia on antibody-dependent dengue virus infection in human monocytes. The authors demonstrate that physiological low oxygen levels, similar to those present in lymphoid organs where dengue virus has been found to replicate, upregulate FcγRIIA expression in human monocytes partially through the hypoxia-inducible factor alpha (HIF-1α), associated with increased uptake of dengue virus immune complexes and production of infectious dengue virus progenies. Furthermore, they report that hypoxia-induced but HIF-1α-independent

alterations in membrane lipid compositions are also necessary for antibody-dependent enhancement of dengue virus infection.

General comments:

Overall, the work tackles an interesting topic and sheds new light on the influence of physiological low levels of oxygen on monocyte susceptibility to antibody-dependent dengue virus infection. The scientific rationale for the investigation is sound and the authors present a few novel and important observations that provide a logical extension of previous work from this team on dengue virus pathogenesis. The manuscript is well organized and carried out. However, some information are missing and should be provided to clarify a few issues, and additional experiments are needed to strengthen the study and further support the authors' conclusions

The following points should be addressed:

Major Criticisms

1. Data significance appears overestimated in some of the graph shown (e.g. Fig. 1e, Fig. 2b,e, Fig. 3c,e...), based on the visual inspection of the images. However, it is very difficult to make a proper evaluation because no information is provided on the number of donors tested (in case of primary monocytes) or of experiment replicates performed (in case of THP-1), nor it is indicated whether data shown represent mean or median of different experiments and whether SD or SE was used. In Fig. 3a-e, data from single donors are presented, but the graphs still show SD or SE. Do they represent different replicates of a single experiment or different experiments performed at different times with monocytes isolated from the same donor? These information should be provided. Furthermore, the use of the unpaired t test, in place of the more appropriate paired t test, should be justified. It is possible that this is the reason why data significance seems overestimated. A paragraph reporting the statistical analyses used to determine data significance is missing and should be included in the Materials and Methods section.
2. The authors state that Fc γ RIIA upregulation by hypoxia is mediated by HIF-1 α , based on data showing that its expression is increased upon monocyte treatment with desferrioxamine (DFX), a known chemical HIF-1 α inducer, under normoxic conditions (Fig. 2b, 2c), whereas it is reduced following HIF-1 α silencing in hypoxic monocytes (2e, 2f). However, only a 30% inhibition of Fc γ RIIA protein expression is achieved upon HIF-1 α silencing, suggesting that hypoxic induction of Fc γ RIIA occurs only in part via HIF-1 α stabilization and that other hypoxia-responsive transcription factors contribute to this effect. In particular, HIF-2 α is known to be stabilized by both hypoxia and DFX in monocytes and to bind HRE to activate gene transcription, similarly to HIF-1 α . Have the authors evaluated HIF-2 α contribution to Fc γ RIIA expression increase? Silencing experiments should be carried out. Furthermore, western blot analysis confirming siRNA-mediated HIF-1 α /-2 α knockdown should be shown. Direct evidence of the effects of HIF-1 α and/or HIF- α silencing on dengue virus internalization or infection could also help strengthening the authors' conclusions.
3. The authors state that dengue virus infection is enhanced in response to hypoxia but not to DFX treatment under normoxia, despite enhanced Fc γ RIIA expression, and conclude that "hypoxia-induced internalization of antibody-opsonized dengue virus likely requires HIF-1 α -independent changes in the cell to complement the induced Fc γ RIIA expression". However, it appears from the results presented in Figure 4g that the percentage of THP-1 cells infected with antibody-opsonized dengue virus significantly increases upon treatment with 200 μ M DFX also after pronase treatment, suggesting enhanced viral entry under normoxic conditions. Evaluation of the effects of stable Fc γ RIIA expression knockdown in DFX-treated monocytes on dengue virus infection would help substantiating or refuting the authors' conclusion.
4. The authors state that alterations in membrane ether-linked PE levels acts synergistically with Fc γ RIIA upregulation to increase uptake of dengue virus immune complexes in hypoxic monocytes. However, no data are presented demonstrating that the effects are synergical. Experiments in which the effects of single and double Fc γ RIIA and AGPS genes silencing on antibody-dependent dengue virus infection are compared would help clarifying this point. Furthermore, assessment of membrane ether-linked PE levels in HIF-silenced cells could be useful to confirm that the observed alterations under hypoxia is independent of HIF activity.

Minor Criticisms:

1. Several mistakes are present in the Reference list, such as wrong date of publication and/or missing volume and page number information. In particular, References by Andrews, Bosco et al. Blood, Capedin et al., Chan et al. Nature Microbiology, Pettersen et al., Villar et al. have to be corrected. Furthermore, title's abbreviations are used for some of the Journals, whereas full length

titles are used for others. Please, carefully revise this section.

2. Some Figure Legends are not fully informative and should be expanded
3. Basal expression of ADM and FcγRIIA mRNA appears quite variable among different figures. These differences should be explained.
4. Why did the authors use βactin for protein data normalization in Fig.4f and 5c instead of Lamp-1 that was used in the other blots? This molecule is a known hypoxia target and it would be preferable not using it for data normalization. Confirmation of findings using Lamp-1 is recommended.
5. How did the authors quantify protein levels assessed by western blot? How many experiments have been carried out to confirm the findings? Please indicate.
6. Data obtained with both primary monocytes and THP-1 cells are shown in Fig EV1. Therefore, the title of the legend should be modified accordingly. I would suggest something like: "exposure to 3% oxygen can upregulate the expression of known hypoxia-inducible genes in human monocytes". Furthermore, the authors should comment in the text why they performed the same experiments using both primary monocytes and THP-1.
7. Results shown in Figures EV2 would be better presented as a regular than a supplementary figure, because it reports critical data.

Referee #3:

The paper by Gan et al is a well written study describing an important issue in DENV biology, namely the role of antibody in DENV entry. In general the results are interesting and compelling, and will be of interest to DENV researchers. However a few issues should be addressed to strengthen the several of the conclusions.

Major points:

- 1) The Western blot data showing FcR in figure 1 protein levels is not very compelling. The reported changes in expression are small, and interpretation of the data is complicated by high background levels. Are these changes consistent? As this change in expression is a major focus of the paper, the authors should provide additional more compelling data. The authors should graph the average change in protein abundance from at least three experiments, and include appropriate statistical analysis to demonstrate significant change.
- 2) The issue with non-specific antibody binding similarly complicates the flow cytometry analysis in Figure 1C. The authors should show that FcγRIIa blocking antibody reduces the observed MFI.
- 3) In figure 2 the authors use siRNA to HIF1 alpha. The data showing the extent of HIF-1alpha depletion should be shown. Similarly, the induction of HIF-1alpha by DFX should be shown.
- 4) In figure 2g, the authors need to include ChiP-Seq data for the same locus at 20% O₂ levels. Without this information it is unclear if the observed peaks are specific to hypoxia. The authors should also include data for this locus using a control antibody.
- 5) Demonstrating the effect of AGPS depletion on in the DENV neutralization assay as in figure 3 would further strengthen the authors' conclusions that specific lipids are required for hypoxia induced DENV entry.

Minor points:

- 1) references should be separated from text by a space
- 2) In figure 1, are the authors certain that TBP mRNA levels are not affected by hypoxia? This is important, as reference gene expression must be constant under all conditions in order for accurate comparison of expression of the test gene under the different conditions.
- 2) In figure 2g, the ChiP-Seq graph should have a labeled x-axis. As it stands it is unclear where the peaks lie in relation to the transcription start site of FcγRIIA.
- 3) The novel HRE-like site sequence should be indicated on the graph, and the sequence should be

shown in comparison to the consensus HRE site to allow the reader to compare.

1st Revision - authors' response

10 January 2017

Referee #1:

The authors studied various cell-based phenomena that occur when primary human monocytes or THP-1 cells infected by dengue viruses under enhancing conditions are maintained at low compared with ambient (atmospheric) oxygen concentrations. They observed increased expression of Fc gamma receptors and an increased dengue 2 infection of these cells in the presence of enhancing concentrations of two monoclonal antibodies. The observed increase in growth of dengue 2 virus was attributed to an increase in uptake of infectious immune complexes. Experimental design was restricted, perhaps because of disagreements with results of others, with the result that the effect of anoxia on the efficiency of completion of the replication and assembly of dengue viruses was not studied in the presence and absence of enhancing antibodies. Anoxia may differentially effect the contribution of innate immunity to ADE.

We thank Reviewer #1 for the suggestion.

In the case of anoxia, we are unable to directly assess the effect of cells cultured under 0% O₂ as monocytes could not survive under such culture conditions. Instead, we have assessed ADE under 0.5% O₂ tensions. The results show that the phenotype observed at conditions close to anoxia remains similar to that at 3% O₂ tensions at 18hpi (Appendix Figure S2a), where increased DENV mRNA was observed at 18hpi under ADE condition (Appendix Figure S2b). Consistent with previous studies (Ng *et al*, 2014), 0.5% O₂ tensions resulted in upregulation of the HIF1 α target gene, vascular endothelial growth factor (VEGF) at 24 hours post-incubation (Appendix Figure S2c). This is now included in the manuscript page 7 lines 138-140 and Appendix Figure S2.

Specific comments:

1. P 3, line 10. Please do not forget infants who acquire the dengue vascular permeability syndrome as the result of infection enhancement from mothers who have experience two or more dengue infections. ADE does not occur simply as the result of one prior dengue infection.

We thank Reviewer #1 for this reminder and have hence inserted this information along with the supporting references into the manuscript page 3 lines 60-63.

2. P 3, line 14. Does the absence of references to other described mechanisms of intrinsic ADE mean the authors reject or simply choose to ignore them? 1,2

We agree that other described mechanisms of intrinsic ADE should be mentioned and added the relevant details and references in page 3 lines 58-60.

3. P3, line 17. "pro-inflammatory response?" Presumably, the authors do not accept data that supports a direct role of dengue NS1 as a component of tissue damage in acute dengue infections?3,4

We thank Reviewer #1 for pointing out this omission. We have now added sentences that frame the current understanding of dengue pathogenesis more holistically, including the recent reports of NS1 as a toxin that induces vascular leakage and inflammatory cytokines in page 3 lines 63-66.

4. P 12, line 3. Who is it that estimates vaccine efficacy as being due to the circulation of dengue antibodies at levels that prevent dengue enhanced infections, in vitro? Please find a sensible rationale for this research.

We apologize for the poorly phrased sentence and have amended the manuscript to reflect that in vitro measurements of neutralizing antibody titers are vital to determine vaccine immunogenicity and not efficacy in page 14 lines 305.

5. P 12, Primary samples. In order to attribute infection to enhancement, it is critically important to test all PBMC donors for dengue antibodies and exclude all those who are positive. PBMC from dengue immune donors will support dengue infection in the absence of added antibodies. Donors from antibody positive donors can be used for studies such as those described here, but dengue infection enhancement cannot be attributed to antibodies. Please see 5

We agree with Reviewer #1 that it is important to determine the serostatus of PBMC donors, but we respectfully submit that this is important if the study aims at defining the difference between DENV only and antibody-opsonized infection. In this part of the study, our goal is not to examine such a difference but our aim is to demonstrate increased infection of antibody-opsonized DENV in monocytes cultured under different oxygen tensions, either 20% or 3% O₂. Monocytes isolated and cultured from the same Donor is split into 20% O₂ and 3% O₂, thus the presence or absence of any antibodies will be similar in both conditions.

6. P 12, Virus infection. What MOI was used to infect cells? No description is provided of methods for study of enhanced dengue infections of primary human monocytes. Figure 3. What time after infection was dengue virus growth performance measured? These studies are performed under enhancing conditions but they do not measure impact on infection enhancement per se which requires a non-antibody control.

We thank Reviewer #1 for highlighting this missing information. Viruses were infected at MOI of 10 in both DENV only and ADE conditions. In Figure 2, DENV growth performance was measured 72 hours post infection. These descriptions have been added to the materials and methods (page 15 line 332-336) and figure legends.

In addition, non-antibody controls for each donor (Fig 2C-F) and for THP-1 cells (Fig 2K) have been added to illustrate the increase in infections DENV production under enhancing conditions.

1. Ubol S, Halstead SB. How Innate Immune Mechanisms Contribute to Antibody-Enhanced Viral Infections. *Clin Vaccine Immunol* 2010; 17(12): 1829-35.
2. Boonnak K, Dambach KM, Donofrio GC, Marovich MA. Cell Type Specificity and Host Genetic Polymorphisms Influence Antibody Dependent Enhancement of Dengue Virus Infection. *J Virol* 2011; 85(4): 1671-83.
3. Beatty RP, Puerta-Guardo H, Killingbeck S, Glasner D, Harris E. Dengue virus non-structural protein 1 triggers endothelial permeability and vascular leak that can be inhibited by anti-NS1 antibodies. *Science Translational Medicine* 2015; 7(304): 304ra141.
4. Modhiran N, Watterson D, Panetta AK, et al. Dengue virus NS1 is a viral toxin that activates cells via TLR4 and disrupts endothelial cell monolayer integrity. *Science Translational Medicine* 2015; 7(304): 304ra142.
5. Halstead SB. Neutralization and antibody dependent enhancement of dengue viruses *Adv Virus Research* 2003; 60: 421-67.

References

Ng KP, Manjeri A, Lee KL, Huang W, Tan SY, Chuah CTH, Poellinger L & Ong ST (2014) Physiologic hypoxia promotes maintenance of CML stem cells despite effective BCR-ABL1 inhibition. *Blood* **123**: 3316–3326

Referee #2:

This study investigates the effects of hypoxia on antibody-dependent dengue virus infection in human monocytes. The authors demonstrate that physiological low oxygen levels, similar to those present in lymphoid organs where dengue virus has been found to replicate, upregulate FcγRIIA expression in human monocytes partially through the hypoxia-inducible factor alpha (HIF-1α), associated with increased uptake of dengue virus immune complexes and production of infectious dengue virus progenies. Furthermore, they report that hypoxia-induced but HIF-1α-independent alterations in membrane lipid compositions are also necessary for antibody-dependent enhancement of dengue virus infection.

General comments:

Overall, the work tackles an interesting topic and sheds new light on the influence of physiological low levels of oxygen on monocyte susceptibility to antibody-dependent dengue virus infection. The scientific rationale for the investigation is sound and the authors present a few novel and important observations that provide a logical extension of previous work from this team on dengue virus pathogenesis. The manuscript is well organized and carried out. However, some information are missing and should be provided to clarify a few issues, and additional experiments are needed to strengthen the study and further support the authors' conclusions

The following points should be addressed:

Major Criticisms

1. Data significance appears overestimated in some of the graph shown (e.g. Fig.1e, Fig.2b,e, Fig.3c,e...), based on the visual inspection of the images. However, it is very difficult to make a proper evaluation because no information is provided on the number of donors tested (in case of primary monocytes) or of experiment replicates performed (in case of THP-1), nor it is indicated whether data shown represent mean or median of different experiments and whether SD or SE was used. In Fig.3a-e, data from single donors are presented, but the graphs still show SD or SE. Do they represent different replicates of a single experiment or different experiments performed at different times with monocytes isolated from the same donor? These information should be provided.

Furthermore, the use of the unpaired t test, in place of the more appropriate paired t test, should be justified. It is possible that this is the reason why data significance seems overestimated. A paragraph reporting the statistical analyses used to determine data significance is missing and should be included in the Materials and Methods section.

We thank Reviewer #2 for the suggestion and we have now included the necessary details in the figure legends and methods. In Fig 1A-C, F,G and Fig. 2A-F, data from a single donor is shown in each individual graph (4 replicates of a single experiment) with SD. In consultation with a statistician, the use of the unpaired t-test was chosen, as each individual donor is considered a population in an analysis. Paired t-test was only applied in Fig 1E where 4 different donor results were included in one analysis and Fig 1J where THP-1 results was replicated in 4 independent experiments. To demonstrate significance, the significance value is displayed using asterisk (* $p < 0.05$; ** $p < 0.01$; *** $p < 0.001$) and a paragraph reporting statistical analysis has been included in the Materials and Methods (page 19-20 lines 433-439) and figure legends under data information.

2. The authors state that Fc γ RIIA upregulation by hypoxia is mediated by HIF-1 α , based on data showing that its expression is increased upon monocyte treatment with desferrioxamine (DFX), a known chemical HIF-1 α inducer, under normoxic conditions (Fig.2b, 2c), whereas it is reduced following HIF-1 α silencing in hypoxic monocytes (2e,2f). However, only a 30% inhibition of Fc γ RIIA protein expression is achieved upon HIF-1 α silencing, suggesting that hypoxic induction of Fc γ RIIA occurs only in part via HIF-1 α stabilization and that other hypoxia-responsive transcription factors contribute to this effect. In particular, HIF-2 α is known to be stabilized by both hypoxia and DFX in monocytes and to bind HRE to activate gene transcription, similarly to HIF-1 α . Have the authors evaluated HIF-2 α contribution to Fc γ RIIA expression increase? Silencing experiments should be carried out. Furthermore, western blot analysis confirming siRNA-mediated HIF-1 α /-2 α knockdown should be shown. Direct evidence of the effects of HIF-1 α and/or HIF- α silencing on dengue virus internalization or infection could also help strengthening the authors' conclusions.

We appreciate this important nuance raised by Reviewer #2. Western blots confirming siRNA knockdown of HIF1 α has been included as Figure 4A. We agree that 30% inhibition of Fc γ RIIA protein expression is modest (Fig 4D) although this level of inhibition is to be expected due to the 40-50% knockdown efficiency (40-50%) of HIF1 α in THP-1 (Fig 4A), which is the level commonly encountered in suspension cells (Kusumawati *et al*, 1999; Ohtani *et al*, 1989).

To directly address the involvement of HIF2 α , we performed siRNA knockdown of HIF2 α (Fig EV2A) but we did not observe a corresponding decrease in Fc γ RIIA expression (Fig EV2B). In addition, silencing HIF2 α had no effect on DENV infection (Fig EV2C) or opsonized DENV infection (Fig EV2D). Therefore we conclude that it is unlikely that HIF2 α contributes to the increase in Fc γ RIIA expression or increased in DENV immune complex infection under hypoxic conditions. These have been included in the manuscript in page 8 lines 180-182.

3. The authors state that dengue virus infection is enhanced in response to hypoxia but not to DFX treatment under normoxia, despite enhanced Fc γ RIIA expression, and conclude that "hypoxia-induced internalization of antibody-opsonized dengue virus likely requires HIF-1 α -independent changes in the cell to complement the induced Fc γ RIIA expression". However, it appears from the results presented in Figure 4g that the percentage of THP-1 cells infected with antibody-opsonized dengue virus significantly increases upon treatment with 200 μ M DFX also after pronase treatment, suggesting enhanced viral entry under normoxic conditions. Evaluation of the effects of stable Fc γ RIIA expression knockdown in DFX-treated monocytes on dengue virus infection would help substantiating or refuting the authors' conclusion.

We thank Reviewer#2 for this observation. As suggested, we have infected stable knockdown of Fc γ RIIA by shRNA under DFX conditions with AF488 labeled antibody-opsonized DENV. Treatment of shControl cells resulted in the increase in immunofluorescence signals but no difference was observed in DFX treated shFc γ RIIA cells (Fig 4I). However, upon knockdown of HIF1 α in hypoxic THP-1 cells, the resultant decrease in Fc γ RIIA only led to a modest decrease in antibody-opsonized DENV infection. This suggests that additional HIF1 α independent factors are required to increase ADE in addition to the increase in Fc γ RIIA.

We thus infected hypoxic, normoxic and DFX treated cells with antibody-opsonized DENV and observed that both hypoxic and DFX treated THP-1 cells showed increased binding, 51% and 47% respectively, as compared to normoxic THP-1 cells. However, after pronase treatment to remove attached but not internalized virus, infection remained high only in hypoxic THP-1 cells (43%) as compared to normoxic THP-1 cells (20%). DFX treated THP-1 cells only showed a marginal increase (25%) compared to normoxic THP-1 cells (20%) (Fig 5D). Therefore, as DFX cells show a significantly lower DENV internalization as compared to hypoxic cells, this finding further suggests that internalization under hypoxic conditions requires another factor to complement the upregulation of Fc γ RIIA.

4. The authors state that alterations in membrane ether-linked PE levels acts synergistically with Fc γ RIIA upregulation to increase uptake of dengue virus immune complexes in hypoxic monocytes. However, no data are presented demonstrating that the effects are synergical. Experiments in which the effects of single and double Fc γ RIIA and AGPS genes silencing on antibody-dependent dengue virus infection are compared would help clarifying this point. Furthermore, assessment of membrane ether-linked PE levels in HIF-silenced cells could be useful to confirm that the observed alterations under hypoxia is independent of HIF activity.

We thank Reviewer #2 for this very helpful suggestion. To address the synergic effects of AGPS and Fc γ RIIA knockdowns, we have performed additional experiments using single and double knockdowns of Fc γ RIIA and AGPS under hypoxic conditions as shown in Fig 7D. The results show that knockdown of either AGPS or Fc γ RIIA (Fig 7D) resulted in a decrease in DENV internalization (Fig 7E). Additionally, double knockdown of both AGPS and Fc γ RIIA further decreased ADE of DENV infection (Fig 7E). Thanks to Reviewer #2's suggestion, this new set of data now adds strength to our conclusion that alterations in membrane ether-linked PE levels acts synergistically with the increase in Fc γ RIIA under hypoxic conditions to increase DENV immune complexes internalization in THP-1 cells. These have been included in Fig 7E and in the manuscript page 11-12 lines 250-252.

Furthermore, as suggested by the reviewer, we have also measured membrane ether-linked PE levels in HIF1 α silenced cells (Fig EV5A) and showed, with the exception of PE(O-34:1)/oddPE 33:1, no significant difference in the concentrations of PE between control and HIF1 α silenced cells (Fig EV5B-D). Hence, HIF1 α upregulates Fc γ RIIA but not membrane ether-linked PE levels. Significance was tested by unpaired t-test. These have been included in Fig EV5 and in the manuscript page 11 lines 239-241.

Minor Criticisms:

1. Several mistakes are present in the Reference list, such as wrong date of publication and/or missing volume and page number information. In particular, References by Andrews, Bosco et al. Blood, Capedin et al., Chan et al. Nature Microbiology, Pettersen et al., Villar et al. have to be corrected. Furthermore, title's abbreviations are used for some of the Journals, whereas full length titles are used for others. Please, carefully revise this section.

We thank Reviewer #2 for pointing out the inconsistencies in the reference list. We have now amended the references according to the reviewer's recommendation.

2. Some Figure Legends are not fully informative and should be expanded

We have expanded the figure legends to include more details including statistics and sample sizes.

3. Basal expression of ADM and Fc γ RIIA mRNA appears quite variable among different figures. These differences should be explained.

We agree that the levels of ADM and Fc γ RIIA mRNA is variable among different figures. For example, basal expression of ADM is higher in Fig 4B as compared to Fig 4F. This however is expected as cells in Fig 2B were cultured under hypoxic conditions and THP-1 cells in Fig 2G were cultured under normoxic conditions. As ADM has been shown to be a HIF1 α target gene in THP-1 cells (Fig EV1D), its basal expression under hypoxic conditions is thus higher than normoxic conditions.

The variability in Fc γ RIIA mRNA in Fig 4C (hypoxia) and Fig 4G (normoxia) supports our hypothesis that Fc γ RIIA is increased under hypoxic conditions.

To emphasize the difference in the 2 conditions, we have color-coded normoxia in blue (Figure 4F,G) and hypoxia (Figure 4B,C) in red.

4. Why did the authors use β actin for protein data normalization in Fig.4f and 5c instead of Lamp-1 that was used in the other blots? This molecule is a known hypoxia target and it would be preferable not using it for data normalization. Confirmation of findings using Lamp-1 is recommended.

We agree with Reviewer #2 that LAMP-1 should be consistently used for easy comparison. We have hence amended Fig 3F and 7A to use LAMP1 as loading controls for normalization.

5. How did the authors quantify protein levels assessed by western blot? How many experiments have been carried out to confirm the findings? Please indicate.

We have quantified the protein levels by ImageJ v4.7 and normalized with loading controls (LAMP1) for all western blots in the manuscript. Each blot is representative of 3 independent experiments carried out to confirm the findings. Details have been included in the Materials and Methods page 17 lines 370-371 and in figure legends.

6. Data obtained with both primary monocytes and THP-1 cells are shown in Fig EV1. Therefore, the title of the legend should be modified accordingly. I would suggest something like: "exposure to 3% oxygen can upregulate the expression of known hypoxia-inducible genes in human monocytes". Furthermore, the authors should comment in the text why they performed the same experiments using both primary monocytes and THP-1.

We thank Reviewer #2 for the suggestion. The title of the legend to Figure EV1 has been modified accordingly. We have performed the same experiment with both THP-1 and primary monocytes to ensure that the hypoxic response data obtained from cell lines were representative of an *ex vivo* system. This is reflected in the manuscript in page 5 lines 107-109.

7. Results shown in Figures EV2 would be better presented as a regular than a supplementary figure, because it reports critical data.

We thank Reviewer #2 for the suggestion. Figure EV2 is now presented in Fig 6A.

References

Kusumawati A, Commes T, Liautard JP & Widada JS (1999) Transfection of myelomonocytic cell lines: cellular response to a lipid-based reagent and electroporation. *Anal. Biochem.* **269**: 219–221

Ohtani K, Nakamura M, Saito S, Nagata K, Sugamura K & Hinuma Y (1989) Electroporation: application to human lymphoid cell lines for stable introduction of a transactivator gene of human T-cell leukemia virus type I. *Nucl. Acids Res.* **17**: 1589–1604

Referee #3:

The paper by Gan et al is a well written study describing an important issue in DENV biology, namely the role of antibody in DENV entry. In general the results are interesting and compelling, and will be of interest to DENV researchers. However a few issues should be addressed to strengthen the several of the conclusions.

Major points:

1) The Western blot data showing FcR in figure 1 protein levels is not very compelling. The reported changes in expression are small, and interpretation of the data is complicated by high background levels. Are these changes consistent? As this change in expression is a major focus of the paper, the authors should provide additional more compelling data. The authors should graph the average change in protein abundance from at least three experiments, and include appropriate statistical analysis to demonstrate significant change.

We thank Reviewer #3 for highlighting this weakness in our original submission. To address this point, we have now quantified the average change in protein abundance of Fc γ R1IA in normoxic and hypoxic conditions in primary monocytes (Fig 1E) and THP-1 cells (Fig 1J) from 4 independent experiments. Upregulation of Fc γ R1IA was observed in both hypoxic primary monocytes and THP-1 cells. Significance was shown by paired t-test. The method of statistical analysis has been included in the material and methods (page 19-20 lines 433-439) and figure legends.

2) The issue with non-specific antibody binding similarly complicates the flow cytometry analysis in Figure 1C. The authors should show that Fc γ R1IA blocking antibody reduces the observed MFI.

We agree with Reviewer #3 that non-specific binding can complicate the flow cytometry analysis. As suggested, we have conducted this experiment again using a goat anti-Fc γ R1IA antibody to block Fc γ R1IA followed by a mouse anti-Fc γ R1IA to probe for Fc γ R1IA. Indeed, using a blocking antibody reduces the observed MFI of Fc γ R1IA (Appendix Fig S1). As a control, isotype antibodies were also used and showed minimal fluorescence (Appendix Fig S1). We thus thank Reviewer #3 for this suggestion that, we believe, has improved the quality of our data.

3) In figure 2 the authors use siRNA to HIF1 alpha. The data showing the extent of HIF-1alpha depletion should be shown. Similarly, the induction of HIF-1alpha by DFX should be shown.

We thank Reviewer #3 for the suggestion and have included western blots and its respective densitometry depicting the extent of HIF1 α depletion after siRNA knockdown and DFX treatments in Figure 4A,E.

4) In figure 2g, the authors need to include ChiP-Seq data for the same locus at 20% O₂ levels. Without this information it is unclear if the observed peaks are specific to hypoxia. The authors should also include data for this locus using a control antibody.

We thank Reviewer #3 for this interesting suggestion. In contrast to 3% O₂ conditions, no DNA was enriched with HIF1 α CHiP under 20% O₂ (Appendix Table S2). This inability to enrich for DNA at 20% O₂ is consistent with the fact that HIF1 α is rapidly degraded under normoxic conditions and is only stabilized and transcriptionally active under hypoxic conditions or DFX treatment (Wang & Semenza, 1993a; 1993b). In addition, we have included additional experiments where we used histone H3 as a positive control and rabbit IgG as a negative control. As expected, while we were able to achieve a good enrichment of DNA using Histone H3 antibodies, rabbit IgG ChiP under 20% O₂ conditions did not enrich for any DNA. This data is now shown in Appendix Table S2 and included in the manuscript on page 9 lines 193-195.

Furthermore, to show if the observed peaks are indeed specific to hypoxia, we treated THP-1 cells with DFX to stabilize HIF1 α under 20% O₂ conditions and were able to identify the identical peak that was observed under 3% O₂ conditions. This data is now included in Fig 5B and in the manuscript page 9 lines 196-198. Collectively, our data suggest that the observed peak is specific to HIF1 α .

5) Demonstrating the effect of AGPS depletion on in the DENV neutralization assay as in figure 3 would further strengthen the authors' conclusions that specific lipids are required for hypoxia induced DENV entry.

We thank Reviewer #3 for this useful suggestion. We have hence performed additional experiments where we knockdown AGPS, and show that knockdown of AGPS resulted in decreased plaque titers under ADE conditions (Fig. 7C). This is in part due to the decrease in DENV immune complex entry as shown in Fig 7B, indicating that PE is indeed required for hypoxia induced DENV entry. This is reflected in the manuscript on page 11 lines 248-250.

Minor points:

1) references should be separated from text by a space

We have edited the references to separate from text by a space.

2) In figure 1, are the authors certain that TBP mRNA levels are no affected by hypoxia? This is important, as reference gene expression must be constant under all conditions in order for accurate comparison of expression of the test gene under the different conditions.

We agree that reference gene expression must be constant under both normoxic and hypoxic conditions. TBP mRNA in our system remains constant in 20% and 3% O₂ as assessed by microarray gene expression followed by quantile normalization. The TBP changes between hypoxic and normoxic TBP levels are now shown in Appendix Table S1.

2) In figure 2g, the ChiP-Seq graph should have a labeled x-axis. As it stands it is unclear where the peaks lie in relation to the transcription start site of FcgRIIA.

We thank Reviewer #3 for the suggestion. We have now clearly labeled the ChiP-seq x-axis above Figure 5A,B with relative positions of Chromosome 1.

3) The novel HRE-like site sequence should be indicated on the graph.

The HRE sequence is indicated on the graph with the symbol # in Figure 5A,B.

References

Wang GL & Semenza GL (1993a) Characterization of hypoxia-inducible factor 1 and regulation of DNA binding activity by hypoxia. *Journal of Biological Chemistry* **268**: 21513–21518
Wang GL & Semenza GL (1993b) Desferrioxamine induces erythropoietin gene expression and hypoxia-inducible factor 1 DNA-binding activity: implications for models of hypoxia signal transduction. *Blood* **82**: 3610–3615

2nd Editorial Decision

06 February 2017

Thank you for submitting your revised manuscript to The EMBO Journal. Your study has now been re-reviewed by the three referees and their comments are provided below.

As you can see, the referees appreciate the introduced changes and are supportive of publication here. They have a few remaining concerns that shouldn't involve too much additional work to sort out.

Looking forward to seeing the final version

REFEREE REPORTS

Referee #1:

This manuscript infers from a single cited research study on mice that in humans, relative anoxia is the default internal environment of all lymph nodes, spleen, liver (?), bone marrow (?). All these tissues are hosts to resident monocytes and macrophages that support antibody dependent enhanced (ADE) dengue virus infections. This means that rather low oxygen levels characterize all tissue sites of dengue ADE infections during all phases but particularly early phases of human dengue infections. Medical science knows much about oxygen tensions in blood during acute dengue infections of humans which are nowhere near the levels reported for mouse tissues. But, shouldn't we know more about human tissue oxygen tensions to support the conclusions so confidently offered here? See abstract and discussion: "increased viral burden associated with secondary DENV infection is antibody-dependent but hypoxia-mediated." This reviewer does not understand the meaning of "hypoxia-mediated" as opposed to "hypoxia-dependent." At any rate, a little humility is indicated. The observations presented are suggestive of a modest infection amplifying effect of hypoxia on several component phenomena of ADE. The correct stance, is to either cite studies that demonstrate low tissue oxygen in human tissues or suggest that such studies be performed before imputing to anoxia a pathogenic effect on human dengue disease.

Specific comments:

P 3, line 50. The reviewer suggests the authors describe the broader pathophysiological features of severe dengue: increased vascular permeability, thrombocytopenia, altered hemostasis, activated complement and increased levels of liver transaminases. Their words, "severe circulatory shock, internal hemorrhage and organ dysfunction" are either rare or not necessarily due to dengue infection (organ dysfunction).

P 3, line 52. "30% mortality" sounds like a "scare-tactic" rather than a considered descriptor.

P 3, line 62. Not "after" but "during"

P 4, line 72. As further evidence of sites of dengue virus infection, the authors may wish to add Aye et al Human Pathology, 2014, a major pathological study of fatal dengue shock syndrome in Myanmar children.

Figure 2. Almost all increases in dengue virus replication measured under high and low oxygen tensions varied by only 2 - 3-fold. While these appear to be reproducible the effect measured is modest.

Referee #2:

The Authors have thoroughly addressed all my concerns and carried out a quite respectable number of additional experiments which strengthen the study and further support its conclusions. The paper has now substantially gained in quality.

However, I still have a remark regarding the statistical analysis used to determine data significance. I don't understand the reason for using the unpaired t test in Figure 1A-C, F,G and Fig. 2A-F, where data from a single donor (4 determinations of a single experiment) are shown, and the paired t-test in Fig 1E and Fig 1J, where 4 different donor/experiments were included in the analysis. In my opinion, paired t test should be applied in all the Figures, because cells from the same donor(s)/experiment(s) are compared after exposure to two different culture conditions. In addition, the paragraph in the Materials and Methods should be rephrased: for example it is not correct to say "Mean of 1 representative experiment and its SD is shown"; better saying "Mean {plus minus} SD of quadruplicate determinations from one representative experiment is shown"

Referee #3:

The revised manuscript addresses each of the concerns noted in the previous review, and is now acceptable for publication.

2nd Revision - authors' response

09 February 2017

Referee #1:

This manuscript infers from a single cited research study on mice that in humans, relative anoxia is the default internal environment of all lymph nodes, spleen, liver (?), bone marrow (?). All these tissues are hosts to resident monocytes and macrophages that support antibody dependent enhanced (ADE) dengue virus infections. This means that rather low oxygen levels characterize all tissue sites of dengue ADE infections during all phases but particularly early phases of human dengue infections. Medical science knows much about oxygen tensions in blood during acute dengue infections of humans which are nowhere near the levels reported for mouse tissues. But, shouldn't we know more about human tissue oxygen tensions to support the conclusions so confidently offered here? See abstract and discussion: "increased viral burden associated with secondary DENV infection is antibody-dependent but hypoxia-mediated." This reviewer does not understand the meaning of "hypoxia-mediated" as opposed to "hypoxia-dependent." At any rate, a little humility is indicated. The observations presented are suggestive of a modest infection amplifying effect of hypoxia on several component phenomena of ADE. The correct stance, is to either cite studies that demonstrate low tissue oxygen in human tissues or suggest that such studies be performed before imputing to anoxia a pathogenic effect on human dengue disease.

We thank Reviewer 1 for his comments. We agree that we do know more with regards to human tissue oxygen tensions. We have now cited a review from Carreau *et al*, 2011 that summarizes, from multiple studies, the normal oxygen partial pressures in human tissues ranging from inspired air in the trachea to the lymphoid organs and bone marrow. With the exception of inspired air in the trachea (19.7% O₂), aveoli (14.5% O₂) and in arterial blood (13.2% O₂), all other organs tissue sites have been shown to have oxygen pressures below 10% O₂ (Carreau *et al*, 2011). While it is true that oxygen tensions in blood during acute dengue infections are higher than that reported for tissue levels, active replication of the virus have been shown to occur in lymphoid organs and the liver, both of which have low oxygen tensions. Moreover, DENV is trafficked to draining lymph nodes probably by dendritic cells from the skin after an infective mosquito bite to establish systemic infection (St John *et al*, 2013). Consequently, the low physiological oxygen levels in lymph nodes, lymphoid organs and the liver are thus important considerations for pathogenesis studies.

We also thank the reviewer for the suggested amendment for using the term hypoxia induced instead of hypoxia dependent. Indeed this phrase captures our conclusions more accurately. We have made all the necessary edits in this revised manuscript.

Specific comments:

P 3, line 50. The reviewer suggests the authors describe the broader pathophysiological features of severe dengue: increased vascular permeability, thrombocytopenia, altered hemostasis, activated complement and increased levels of liver transaminases. Their words, "severe circulatory shock, internal hemorrhage and organ dysfunction" are either rare or not necessarily due to dengue infection (organ dysfunction).

We agree that there is a broad range of outcomes of severe dengue. In addition to what the reviewer has listed, there are also CNS manifestations of severe dengue. However, to detail all of these outcomes would we believe unnecessarily lengthen the outcome of the manuscripts and distract readers from the main purpose of this study, which is to define a hitherto neglected mechanism that contributes to pathogenesis. Our description of severe dengue is consistent with the 2009 World Health Organization classification of severe dengue and has now been added to our references. (page 3 line 52)

P 3, line 52. "30% mortality" sounds like a "scare-tactic" rather than a considered descriptor.

As previously published, mortality rates for severe dengue, if not properly managed, can be as high as 30% (Nimmannitya, 1997; Ooi *et al*, 2006). We therefore disagree with Reviewer 1 that this mortality rate constitutes a scare tactic but is instead a reported clinical observation. This citation has been included in the manuscript (page 3 line 53)

P 3, line 62. Not "after" but "during"

We thank Reviewer 1 for the correction and have amended the manuscript accordingly. (page 3 line 62)

*P 4, line 72. As further evidence of sites of dengue virus infection, the authors may wish to add Aye *et al* Human Pathology, 2014, a major pathological study of fatal dengue shock syndrome in Myanmar children.*

We thank Reviewer 1 for the suggestion and have included Aye *et al* to the reference list. (page 4 line 72)

Figure 2. Almost all increases in dengue virus replication measured under high and low oxygen tensions varied by only 2 - 3-fold. While these appear to be reproducible the effect measured is modest.

We agree that DENV replication measured under 20% and 3% oxygen tensions varied by 2-3 fold. This is however the result of only one cycle of infection and replication. The incubation period of dengue ranges from 3 to 14 days. If each cycle of virus replication takes approximately 24h, as it does in in vitro systems, a 2-3 fold difference from each cycle would translate to 6-42 fold difference in viral burden at the onset of symptoms. We believe this difference is not trivial. Moreover, the thrust of this paper is not just to highlight the difference in outcome of viral replication, but rather the hypoxia induced mechanisms that until now have not been examined as factors that contribute to dengue pathogenesis. It is likely that there are a plethora of hypoxia-induced effects that contribute to both efficiency of infection and disease manifestation, all of which will require detailed investigations. This paper therefore provides the basis for these studies to be conducted expediently.

References

- Carreau A, Hafny Rahbi BE, Matejuk A, Grillon C & Kieda C (2011) Why is the partial oxygen pressure of human tissues a crucial parameter? Small molecules and hypoxia. *J. Cell. Mol. Med.* **15**: 1239–1253
- Nimmannitya S (1997) Dengue hemorrhagic fever: diagnosis and management. In: Dengue and dengue hemorrhagic fever. p133-145. Oxford: CABI Publishing
- Ooi EE, Goh K-T & Gubler DJ (2006) Dengue Prevention and 35 Years of Vector Control in Singapore. *Emerg. Infect. Dis.* **12**: 887–893
- St John AL, Abraham SN & Gubler DJ (2013) Barriers to preclinical investigations of anti-dengue immunity and dengue pathogenesis. *Nat. Rev. Microbiol.* **11**: 420–426

Referee #2:

The Authors have thoroughly addressed all my concerns and carried out a quite respectable number of additional experiments which strengthen the study and further support its conclusions. The paper has now substantially gained in quality.

However, I still have a remark regarding the statistical analysis used to determine data significance. I don't understand the reason for using the unpaired t test in Figure 1A-C, F,G and Fig. 2A-F, where data from a single donor (4 determinations of a single experiment) are shown, and the paired t-test in Fig 1E and Fig 1J, where 4 different donor/experiments were included in the analysis. In my opinion, paired t test should be applied in all the Figures, because cells from the same donor(s)/experiment(s) are compared after exposure to two different culture conditions. In addition, the paragraph in the Materials and Methods should be rephrased: for example it is not correct to

say "Mean of 1 representative experiment and its SD is shown"; better saying "Mean {plus minus} SD of quadruplicate determinations from one representative experiment is shown"

We thank Reviewer 2 for the observation, however after consultation with our statistician, Prof Yin Bun Cheung, the use of paired and unpaired t-test in our figures is appropriate.

The data structure in Fig 1E and 1J is not comparable to that in Fig 1A-C,F,G and Fig 2A-F. In Fig 1E, each of the four donor's cells are cultured under two conditions, giving totally 8 data points (4 in each condition). Within each condition, the variations between the 4 data points are due to differences between donors (in addition to pure noise). Additionally, there is a rationale to form pairs of data points: the same donor's data points in the two conditions form a pair. As such, the paired t-test is appropriate because it takes the variation between donors (that is variation not due to culture conditions) into account and is operational (because the pairing is not arbitrary). Similarly in Fig 1J, THP-1 cells were cultured under 2 conditions, giving in total 2 data points (1 in each condition). This experiment was then repeated 4 independent times with different passages of THP-1 cells. As such, the paired t-test is appropriate because it takes into account the variation between each experiment. Additionally, the pairing is not arbitrary as data points in 20% conditions were paired to 3% conditions performed at the same time.

In contrast, in Fig 1A-C,F,G and Fig 2A-F, the same donor's cells are divided into 8 replicates, 4 of which placed under 20% O₂ and 4 of which placed under 3% O₂. The variation between the 4 data points under the same condition is pure noise. There is no rationale to say which data point from condition 1 should go with which data point from condition 2 to form a pair. Any attempt to form 4 pairs of data points is arbitrary. As such, the paired t-test is not only inappropriate (because the only systematic variation is from the conditions) but also not operational (because there is no logical way to pair the data points).

The section of materials and methods has been amended accordingly as suggested. (page 19 line 435-436)

To the editor: we have included the p-values if we used paired vs unpaired t-test for each data point (for figures which reviewer 2 has asked for paired t-test). In either case the results are still significant.

Figure	p-value (unpaired t-test)	p-value (paired t-test)
1A	6h: 0.03 12h: 0.01 18h: 0.019 24h: 0.002	6h: 0.01 12h: 0.02 18h: 0.03 24h: 0.03
1B	6h: 0.04 12h: 0.03 18h: 0.004 24h: 0.01	6h: 0.0093 12h: 0.004 18h: 0.01 24h: 0.04
1C	6h: 0.07 (non significant) 12h: 0.004 18h: 0.001 24h: 0.02	6h: 0.1 (non-significant) 12h: 0.005 18h: 0.01 24h: 0.01
1F	0.001	0.006
1G	0.8	0.87
2A	<0.0001 (peak enhancement)	0.002 (peak enhancement)
2B	<0.0001 (peak enhancement)	0.002 (peak enhancement)
2C	<0.0001	<0.0001
2D	0.0001	0.014
2E	<0.0001	0.0005
2F	0.0008	0.001

Referee #3:

The revised manuscript addresses each of the concerns noted in the previous review, and is now acceptable for publication.

Corresponding Author Name: Dr Ooi Eng Eong

Journal Submitted to: EMBO

Manuscript Number: EMBOJ-2016-95642